# Towards Optimal Off-Policy Evaluation for Reinforcement Learning with Marginalized Importance Sampling

**Tengyang Xie**[*]
Dept. of Computer Science
UIUC
Urbana, IL 61801
tx10@illinois.edu

**Yifei Ma**
AWS AI Labs
Amazon.com Services, Inc.
East Palo Alto, CA 94303
yifeim@amazon.com

**Yu-Xiang Wang**
Dept. of Computer Science,
UC Santa Barbara
Santa Barbara, CA 93106
yuxiangw@cs.ucsb.edu

## Abstract

Motivated by the many real-world applications of reinforcement learning (RL) that require safe-policy iterations, we consider the problem of off-policy evaluation (OPE) — the problem of evaluating a new policy using the historical data obtained by different behavior policies — under the model of nonstationary episodic Markov Decision Processes (MDP) with a long horizon and a large action space. Existing importance sampling (IS) methods often suffer from large variance that depends exponentially on the RL horizon $H$. To solve this problem, we consider a marginalized importance sampling (MIS) estimator that recursively estimates the state marginal distribution for the target policy at every step. MIS achieves a mean-squared error of

$$\frac{1}{n} \sum\nolimits_{t=1}^{H} \mathbb{E}_\mu \left[ \frac{d_t^\pi(s_t)^2}{d_t^\mu(s_t)^2} \mathrm{Var}_\mu \left[ \frac{\pi_t(a_t|s_t)}{\mu_t(a_t|s_t)} \big( V_{t+1}^\pi(s_{t+1}) + r_t \big) \Big| s_t \right] \right] + \tilde{O}(n^{-1.5})$$

where $\mu$ and $\pi$ are the logging and target policies, $d_t^\mu(s_t)$ and $d_t^\pi(s_t)$ are the marginal distribution of the state at $t$th step, $H$ is the horizon, $n$ is the sample size and $V_{t+1}^\pi$ is the value function of the MDP under $\pi$. The result matches the Cramer-Rao lower bound in Jiang and Li [2016] up to a multiplicative factor of $H$. To the best of our knowledge, this is the first OPE estimation error bound with a polynomial dependence on $H$. Besides theory, we show empirical superiority of our method in time-varying, partially observable, and long-horizon RL environments.

## 1 Introduction

The problem of *off-policy evaluation* (OPE), which predicts the performance of a policy with data only sampled by a behavior policy [Sutton and Barto, 1998], is crucial for using *reinforcement learning* (RL) algorithms responsibly in many real-world applications. In many settings where RL algorithms have already been deployed, e.g., targeted advertising and marketing [Bottou et al., 2013; Tang et al., 2013; Chapelle et al., 2015; Theocharous et al., 2015; Thomas et al., 2017] or medical treatments [Murphy et al., 2001; Ernst et al., 2006; Raghu et al., 2017], online policy evaluation is usually expensive, risky, or even unethical. Also, using a bad policy in these applications is dangerous and could lead to severe consequences. Solving OPE is often the starting point in many RL applications.

To tackle the problem of OPE, the idea of importance sampling (IS) corrects the mismatch in the distributions under the behavior policy and target policy. It also provides typically unbiased or

---

[*]The research was partially conducted when TX was visiting YW and YM at Amazon AWS AI Labs during his internship in Summer 2018.

strongly consistent estimators [Precup et al., 2000]. IS-based off-policy evaluation methods have also seen lots of interest recently especially for short-horizon problems, including contextual bandits [Murphy et al., 2001; Hirano et al., 2003; Dudík et al., 2011; Wang et al., 2017]. However, the variance of IS-based approaches [Precup et al., 2000; Thomas et al., 2015; Jiang and Li, 2016; Thomas and Brunskill, 2016; Guo et al., 2017; Farajtabar et al., 2018] tends to be too high to provide informative results, for long-horizon problems [Mandel et al., 2014], since the variance of the product of importance weights may grow exponentially as the horizon goes long. There are also model-based approaches for solving OPE problems [Liu et al., 2018b; Gottesman et al., 2019], where the value of the target policy is estimated directly using the approximated MDP.

Given this high-variance issue, it is necessary to find an IS-based approach without relying heavily on the cumulative product of importance weights from the whole trajectories. While the benefit of cumulative products is to allow unbiased estimation even without any state observability assumptions, reweighing the entire trajectories may not be necessary if some intermediate states are directly observable. For the latter, based on Markov independence assumptions, we can aggregate all trajectories that share the same state transition patterns to directly estimate the state distribution shifts after the change of policies from the behavioral to the target. We call this approach marginalized importance sampling (MIS), because it computes the *marginal* state distribution shifts at every single step, instead of the product of policy weights.

Related work [Liu et al., 2018a] tackles the high variance issue due to the cumulative product of importance weights. They apply importance sampling on the average visitation distribution of state-action pairs, based on an estimation of the mixed state distribution. Hallak and Mannor [2017] and Gelada and Bellemare [2019] also leverage the same fact in time-invariant MDPs, where they use the stationary ratio of state-action pairs to replace the trajectory weights. However, these methods may not directly work in finite-horizon MDPs, where the state distributions may not mix.

In contrast to the prior work, the first goal of our paper is to study the sample complexity and optimality of the marginalized approach. Specifically, we provide the first finite sample error bound on the mean-square error for our MIS off-policy evaluation estimator under the episodic tabular MDP setting (with potentially continuous action space). Our MSE bound is the exact calculation up to low order terms. Comparing to the Cramer-Rao lower bound established in [Jiang and Li, 2016, Theorem 3] for DAG-MDP, our bound is larger by at most a factor of $H$ and we have good reasons to believe that this additional factor is required for any OPE estimators in this setting.

In addition to the theoretical results, we empirically evaluate our estimator against a number of strong baselines from prior work in a number of time-invariant/time-varying, fully observable/partially observable, and long-horizon environments. Our approach can also be used in most of OPE estimators that leverage IS-based estimators, such as doubly robust [Jiang and Li, 2016], MAGIC [Thomas and Brunskill, 2016], MRDR [Farajtabar et al., 2018] under mild assumptions (Markov assumption).

Here is a road map for the rest of the paper. Section 2 provides the preliminaries of the problem of off-policy evaluation. In Section 3, we offer the design of our marginalized estimator, and we study its information-theoretical optimality in Section 4. We present the empirical results in a number of RL tasks in Section 5. At last, Section 6 concludes the paper.

## 2  Problem formulation

**Symbols and notations.** We consider the problem of off-policy evaluation for a finite horizon, nonstationary, episodic MDP, which is a tuple defined by $M = (\mathcal{S}, \mathcal{A}, T, r, H)$, where $\mathcal{S}$ is the state space, $\mathcal{A}$ is the action space, $T_t : \mathcal{S} \times \mathcal{A} \times \mathcal{S} \to [0, 1]$ is the *transition function* with $T_t(s'|s, a)$ defined by probability of achieving state $s'$ after taking action $a$ in state $s$ at time $t$, and $r_t : \mathcal{S} \times \mathcal{A} \times \mathcal{S} \to \mathbb{R}$ is the expected reward function with $r_t(s, a, s')$ defined by the mean of immediate received reward after taking action $a$ in state $s$ and transitioning into $s'$, and $H$ denotes the finite horizon. We use $\mathbb{P}[E]$ to denote the probability of an event $E$ and $p(x)$ the p.m.f. (or pdf) of the random variable $X$ taking value $x$. $\mathbb{E}[\cdot]$ and $\mathbb{E}[\cdot|E]$ denotes the expectation and conditional expectation given $E$, respectively.

Let $\mu, \pi : \mathcal{S} \to \mathbb{P}_{\mathcal{A}}$ be policies which output a distribution of actions given an observed state. We call $\mu$ the behavioral policy and $\pi$ the target policy. For notation convenience we denote $\mu(a_t|s_t)$ and $\pi(a_t|s_t)$ the p.m.f of actions given state at time $t$. The expectation operators in this paper will either be indexed with $\pi$ or $\mu$, which denotes that all random variables coming from roll-outs from

the specified policy. Moreover, we denote $d_t^\mu(s_t)$ and $d_t^\pi(s_t)$ the induced state distribution at time $t$. When $t = 1$, the initial distributions are identical $d_1^\mu = d_1^\pi = d_1$. For $t > 1$, $d_t^\mu(s_t)$ and $d_t^\pi(s_t)$ are functions of not just the policies themselves but also the unknown underlying transition dynamics, i.e., for $\pi$ (and similarly $\mu$), recursively define

$$d_t^\pi(s_t) = \sum_{s_{t-1}} P_t^\pi(s_t|s_{t-1})d_{t-1}^\pi(s_{t-1}),$$

$$\text{where } P_t^\pi(s_t|s_{t-1}) = \sum_{a_{t-1}} T_t(s_t|s_{t-1}, a_{t-1})\pi(a_{t-1}|s_{t-1}). \tag{2.1}$$

We denote $P_{i,j}^\pi \in \mathbb{R}^{S \times S}$ $\forall j < i$ as the state-transition probability from step $j$ to step $i$ under a sequence of actions taken by $\pi$. Note that $P_{t+1,t}^\pi(s'|s) = \sum_a P_{t+1,t}(s'|s, a)\pi_t(a|s) = T_{t+1}(s'|s, \pi_t(s))$.

Behavior policy $\mu$ is used to collect data in the form of $(s_t^{(i)}, a_t^{(i)}, r_t^{(i)}) \in \mathcal{S} \times \mathcal{A} \times \mathbb{R}$ for time index $t = 1, \ldots, H$ and episode index $i = 1, \ldots, n$. Target policy $\pi$ is what we are interested to evaluate. Also, let $\mathcal{D}$ to denote the historical data, which contains $n$ episode trajectories in total. We also define $\mathcal{D}_h = \{(s_t^{(i)}, a_t^{(i)}, r_t^{(i)}) : i \in [n], t \le h\}$ to be roll-in realization of $n$ trajectories up to step $h$.

Throughout the paper, probability distributions are often used in their vector or matrix form. For instance, $d_t^\pi$ without an input is interpreted as a vector in a $S$-dimensional probability simplex and $P_{i,j}^\pi$ is then a stochastic transition matrix. This allows us to write (2.1) concisely as $d_{t+1}^\pi = P_{t+1,t}^\pi d_t^\pi$.

Also note that while $s_t, a_t, r_t$ are usually used to denote fixed elements in set $\mathcal{S}, \mathcal{A}$ and $\mathbb{R}$, in some cases we also overload them to denote generic random variables $s_t^{(1)}, a_t^{(1)}, r_t^{(1)}$. For example, $\mathbb{E}_\pi[r_t] = \mathbb{E}_\pi[r_t^{(1)}] = \sum_{s_t,a_t,s_{t+1}} d^\pi(s_t, a_t, s_{t+1})r_t(s_t, a_t, s_{t+1})$ and $\text{Var}_\pi[r_t(s_t, a_t, s_{t+1})] = \text{Var}_\pi[r_t(s_t^{(1)}, a_t^{(1)}, s_{t+1}^{(1)})]$. The distinctions will be clear in each context.

**Problem setup.** The problem of off-policy evaluation is about finding an estimator $\widehat{v}^\pi : (\mathcal{S} \times \mathcal{A} \times \mathbb{R})^{H \times n} \to \mathbb{R}$ that makes use of the data collected by running $\mu$ to estimate

$$v^\pi = \sum_{t=1}^H \sum_{s_t} d_t^\pi(s_t) \sum_{a_t} \pi(a_t|s_t) \sum_{s_{t+1}} T_t(s_{t+1}|s_t, a_t)r_t(s_t, a_t, s_{t+1}), \tag{2.2}$$

where we assume knowledge about $\mu(a|s)$ and $\pi(a|s)$ for all $(s, a) \in \mathcal{S} \times \mathcal{A}$, but *do not observe* $r_t(s_t, a_t, s_{t+1})$ for any actions other than a noisy version of it the evaluated actions. Nor do we observe the state distributions $d_t^\pi(s_t) \forall t > 1$ implied by the change of policies. Nonetheless, our goal is to find an estimator to minimize the mean-square error (MSE): $\text{MSE}(\pi, \mu, M) = \mathbb{E}_\mu[(\hat{v}^\pi - v^\pi)^2]$, using the observed data and the known action probabilities. Different from previous studies, we focus on the case where $S$ is sufficiently small but $S^2A$ is too large for a reasonable sample size. In other words, this is a setting where we do not have enough data points to estimate the state-action-state transition dynamics, but we do observe the states and can estimate the distribution of the states after the change of policies, which is our main strategy.

**Assumptions:** We list the technical assumptions we need and provide necessary justification.

A1. $\exists R_{\max}, \sigma < +\infty$ such that $0 \le \mathbb{E}[r_t|s_t, a_t, s_{t+1}] \le R_{\max}, \text{Var}[r_t|s_t, a_t, s_{t+1}] \le \sigma^2$ for all $t, s_t, a_t$.

A2. Behavior policy $\mu$ obeys that $d_m := \min_{t,s_t} d_t^\mu(s_t) > 0$ $\forall t, s_t$ such that $d_t^\pi(s_t) > 0$.

A3. Bounded weights: $\tau_s := \max_{t,s} \frac{d_t^\pi(s_t)}{d_t^\mu(s_t)} < +\infty$ and $\tau_a := \max_{t,s_t,a_t} \frac{\pi(a_t|s_t)}{\mu(a_t|s_t)} < +\infty$.

Assumption A1 is assumed without loss of generality. The $\sigma$ bound is required even for on-policy evaluation and the assumption on the non-negativity and $R_{\max}$ can always be obtained by shifting and rescaling the problem. Assumption A2 is necessary for any consistent off-policy evaluation estimator. Assumption A3 is also necessary for discrete state and actions, as otherwise the second moments of the importance weight would be unbounded. For continuous actions, $\tau_a < +\infty$ is stronger than we need and should be considered a simplifying assumption for the clarity of our presentation. Finally, we comment that the dependence in the parameter $d_m, \tau_s, \tau_a$ do not occur in the leading $O(1/n)$ term of our MSE bound, but only in simplified results after relaxation.

# 3   Marginalized Importance Sampling Estimators for OPE

In this section, we present the design of marginalized IS estimators for OPE. For small action spaces, we may directly build models by the estimated transition function $T_t(s_t|s_{t-1}, a_{t-1})$ and the reward function $r_t(s_t, a_t, s_{t+1})$ from empirical data. However, the models may be inaccurate in large action spaces, where not all actions are frequently visited. Function approximation in the models may cause additional biases from covariate shifts due to the change of policies. Standard importance sampling estimators (including the doubly robust versions)[Dudík et al., 2011; Jiang and Li, 2016] avoid the need to estimate the model's dynamics but rather directly approximating the expected reward:

$$\widehat{v}_{\text{IS}}^{\pi} = \frac{1}{n} \sum_{i=1}^{n} \sum_{h=1}^{H} \left[ \prod_{t=1}^{h} \frac{\pi(a_t^{(i)}|s_t^{(i)})}{\mu(a_t^{(i)}|s_t^{(i)})} \right] r_h^{(i)}.$$

To adjust for the differences in the policy, importance weights are used and it can be shown that this is an unbiased estimator of $v^{\pi}$ (See more detailed discussion of IS and the doubly robust version in Appendix C). The main issue of this approach, when applying to the episodic MDP with large action space is that the variance of the importance weights grows exponentially in $H$ [Liu et al., 2018a], which makes the sample complexity exponentially worse than the model-based approaches, when they are applicable. We address this problem by proposing an alternative way of estimating the importance weights which achieves the same sample complexity as the model-based approaches while allowing us to achieve the same flexibility and interpretability as the IS estimator that does not explicitly require estimating the state-action dynamics $T_t$. We propose the Marginalized Importance Sampling (MIS) estimator:

$$\widehat{v}_{\text{MIS}}^{\pi} = \frac{1}{n} \sum_{i=1}^{n} \sum_{t=1}^{H} \frac{\widehat{d}_t^{\pi}(s_t^{(i)})}{\widehat{d}_t^{\mu}(s_t^{(i)})} \widehat{r}_t^{\pi}(s_t^{(i)}). \tag{3.1}$$

Clearly, if $\widehat{d}^{\pi} \to d_t^{\pi}, \widehat{d}^{\mu} \to d_t^{\mu}, \widehat{r}_t^{\pi} \to \mathbb{E}_{\pi}[R_t(s_t, a_t)|s_t]$, then $\widehat{v}_{\text{MIS}}^{\pi} \to v^{\pi}$.

It turns out that if we take $\widehat{d}_t^{\mu}(s_t) := \frac{1}{n} \sum_i \mathbf{1}(s_t^{(i)} = s_t)$ — the empirical mean — and define $\widehat{d}_t^{\pi}(s_t)/\widehat{d}_t^{\mu}(s_t) = 0$ whenever $n_{s_t} = 0$, then (3.1) is equivalent to $\sum_{t=1}^{H} \sum_{s_t} \widehat{d}_t^{\pi}(s_t)\widehat{r}^{\pi}(s_t)$ – the direct plug-in estimator of (2.2). It remains to specify $\widehat{d}_t^{\pi}(s_t)$ and $\widehat{r}^{\pi}(s_t)$. $\widehat{d}_t^{\pi}(s_t)$ is estimated recursively using

$$\widehat{d}_t^{\pi} = \widehat{P}_t^{\pi} \widehat{d}_{t-1}^{\pi}, \text{ where } \widehat{P}_t^{\pi}(s_t|s_{t-1}) = \frac{1}{n_{s_{t-1}}} \sum_{i=1}^{n} \frac{\pi(a_{t-1}^{(i)}|s_{t-1})}{\mu(a_{t-1}^{(i)}|s_{t-1})} \mathbf{1}((s_{t-1}^{(i)}, s_t^{(i)}) = (s_{t-1}, s_t));$$

$$\text{and } \widehat{r}_t^{\pi}(s_t) = \frac{1}{n_{s_t}} \sum_{i=1}^{n} \frac{\pi(a_t^{(i)}|s_t)}{\mu(a_t^{(i)}|s_t)} r_t^{(i)} \mathbf{1}(s_t^{(i)} = s_t), \tag{3.2}$$

where $n_{s_\tau}$ is the empirical visitation frequency to state $s_\tau$ at time $\tau$. Note that our estimator of $r_t^{\pi}(s_t)$ is the standard IS estimators we use in bandits [Li et al., 2015], which are shown to be optimal (up to a universal constant) when $A$ is large [Wang et al., 2017].

The advantage of MIS over the naive IS estimator is that the variance of the importance weight need not depend exponentially in $H$. A major theoretical contribution of this paper is to formalize this argument by characterizing the dependence on $\pi, \mu$ as well as parameters of the MDP $M$. Note that MIS estimator does not dominate the IS estimator. In the more general setting when the state is given by the entire history of observations, Jiang and Li [2016] establishes that no estimators can achieve polynomial dependence in $H$. We give a concrete example later (Example 1) about how IS estimator suffers from the "curse of horizon" [Liu et al., 2018a]. MIS estimator can be thought of as one that exploits the state-observability while retaining properties of the IS estimators to tackle the problem of large action space. As we illustrate in the experiments, MIS estimator can be modified to naturally handle *partially observed* states, e.g., when $s$ is only observed every other step.

Finally, when available, model-based approaches can be combined into importance-weighted methods [Jiang and Li, 2016; Thomas and Brunskill, 2016]. We defer discussions about these extensions in Appendix C to stay focused on the scenarios where model-based approaches are not applicable.

# 4 Theoretical Analysis of the MIS Estimator

Motivated by the challenge of curse of horizon with naive IS estimators, similar to [Liu et al., 2018a], we show that the sample complexity of our MIS estimator reduces to $O(H^3)$. To the best of our knowledge, this is first sample complexity guarantee under this setting, which also matches the Cramer-Rao lower bound for DAG-MDP [Jiang and Li, 2016] as $n \to \infty$ up to a factor of $H$.

**Example 1** (Curse of horizon). *Assume a MDP with i.i.d. state transition models over time and assume that $\frac{\pi_t}{\mu_t}$ is bounded from both sides for all $t$. Suppose the reward is a constant $1$ only shown at the last step, such that naive IS becomes $\widehat{v}_{\mathrm{IS}}^{\pi} = \frac{1}{n} \sum_{i=1}^{n} \left[ \prod_{t=1}^{H} \frac{\pi(a_t^{(i)}|s_t^{(i)})}{\mu(a_t^{(i)}|s_t^{(i)})} \right]$. For every trajectory, $\prod_{t=1}^{H} \frac{\pi_t}{\mu_t} = \exp\left[ \sum_{t=1}^{H} \log \frac{\pi_t}{\mu_t} \right]$; let $E_{\log} = \mathbb{E}[\log \frac{\pi_t}{\mu_t}]$ and $V_{\log} = \mathrm{Var}[\log \frac{\pi_t}{\mu_t}]$. By Central Limit Theorem, $\sum_{t=1}^{H} \log \frac{\pi_t}{\mu_t}$ asymptotically follows a normal distribution with parameters $\left(-HE_{\log}, HV_{\log}\right)$. In other words, $\prod_{t=1}^{H} \frac{\pi_t}{\mu_t}$ asymptotically follows $\mathrm{LogNormal}\left(-HE_{\log}, HV_{\log}\right)$, whose variance is exponential in horizon: $\left(\exp\left(HV_{\log}\right) - 1\right)$. On the other hand, MIS estimates the state distributions recursively, yielding variance that is polynomial in horizon and small OPE errors.*

We now formalize the sample complexity bound in Theorem 4.1.

**Theorem 4.1.** *Let the value function under $\pi$ be defined as follows:*

$$
V_h^{\pi}(s_h) := \mathbb{E}_{\pi} \left[ \sum_{t=h}^{H} r_t(s_t^{(1)}, a_t^{(1)}, s_{t+1}^{(1)}) \middle| s_h^{(1)} = s_h \right] \in [0, V_{\max}], \ \forall h \in \{1, 2, ..., H\}.
$$

*For the simplicity of the statement, define boundary conditions: $r_0(s_0) \equiv 0, \sigma_0(s_0, a_0) \equiv 0, \frac{d_0^{\pi}(s_0)}{d_0^{\mu}(s_0)} \equiv 1, \frac{\pi(a_0|s_0)}{\mu(a_0|s_0)} \equiv 1$ and $V_{H+1}^{\pi} \equiv 0$. Moreover, let $\tau_a := \max_{t,s_t,a_t} \frac{\pi(a_t|s_t)}{\mu(a_t|s_t)}$ and $\tau_s := \max_{t,s_t} \frac{d_t^{\pi}(s_t)}{d_t^{\mu}(s_t)}$. If the number of episodes $n$ obeys that*

$$
n > \max \left\{ \frac{16 \log n}{\min_{t,s_t} d_t^{\mu}(s_t)}, \frac{4t\tau_a\tau_s}{\min_{t,s_t} \max\{d_t^{\pi}(s_t), d_t^{\mu}(s_t)\}} \right\}
$$

*for all $t = 2, ..., H$, then the our estimator $\widehat{v}_{\mathrm{MIS}}^{\pi}$ with an additional clipping step obeys that*

$$
\mathbb{E}[(\mathcal{P}\widehat{v}_{\mathrm{MIS}}^{\pi} - v^{\pi})^2] \leq \frac{1}{n} \sum_{h=0}^{H} \sum_{s_h} \frac{d_h^{\pi}(s_h)^2}{d_h^{\mu}(s_h)} \mathrm{Var}_{\mu} \left[ \frac{\pi(a_h^{(1)}|s_h)}{\mu(a_h^{(1)}|s_h)} (V_{h+1}^{\pi}(s_{h+1}^{(1)}) + r_h^{(1)}) \middle| s_h^{(1)} = s_h \right]
$$

$$
\cdot \left(1 + \sqrt{\frac{16 \log n}{n \min_{t,s_t} d_t^{\mu}(s_t)}}\right) + \frac{19\tau_a^2\tau_s^2 SH^2(\sigma^2 + R_{\max}^2 + V_{\max}^2)}{n^2}.
$$

**Corollary 1.** *In the familiar setting when $V_{\max} = HR_{\max}$, then the same conditions in Theorem 4.1 implies that:*

$$
\mathbb{E}[(\mathcal{P}\widehat{v}_{\mathrm{MIS}}^{\pi} - v^{\pi})^2] \leq \frac{4}{n} \tau_a \tau_s (H\sigma^2 + H^3 R_{\max}^2).
$$

We make a few remarks about the results in Theorem 4.1.

**Dependence on $S$, $A$ and the weights.** The leading term in the variance bound very precisely calculates the MSE of a clipped version of our estimator $\widehat{v}_{\mathrm{MIS}}$[1] modulo a $(1 + O(n^{-1/2}))$ multiplicative factor and an $O(1/n^2)$ additive factor. Specifically, our bound does not explicitly depend on $S$ and $A$ but instead on how similar $\pi$ and $\mu$ are. This allows the method to handle the case when the action space is continuous. The dependence on $\tau_a, \tau_s$ only appear in the low-order terms, while the leading term depends only on the second moments of the importance weights.

**Dependence on $H$.** In general, our sample complexity upper bound is proportional to $H^3$, as Corollary 1 indicates. Our bound reveals that in several cases it is possible to achieve a smaller

exponent on $H$ for specific triplets of $(M, \pi, \mu)$. For instance, when $\pi \approx \mu$, such that $\tau_a, \tau_s = 1 + O(1/H)$, the variance bound gives $O((V_{\max}^2 + H\sigma^2)/n)$ or $O((H^2 R_{\max}^2 + H\sigma^2)/n)$, which matches the MSE bound (up to a constant) of the simple-averaging estimator that knows $\pi = \mu$ a-priori. (See Remark 3 in the Appendix for more details). If $V_{\max}$ is a constant that does not depend on $H$ (this is often the case in games when there is a fixed reward at the end), then the sample complexity is only $O(H)$.

**Optimality.** Comparing to the Cramer-Rao lower bound of the Theorem 3 in [Jiang and Li, 2016], which we paraphrase below

$$\frac{1}{n} \sum_{h=1}^{H} \sum_{s_h} \frac{d_h^\pi(s_h)^2}{d_h^\mu(s_h)} \sum_{a_h} \frac{\pi_h(a_h|s_h)^2}{\mu_h(a_h|s_h)} \text{Var}\left[V_{h+1}^\pi(s_{h+1}^{(1)}) + r_h^{(1)}\Big|s_h^{(1)} = s_h, a_h^{(1)} = a_h\right], \qquad (4.1)$$

the MSE of our estimator is asymptotically bigger by an additive factor of

$$\frac{1}{n} \sum_{h=1}^{H} \sum_{s_h} \frac{d_h^\pi(s_h)^2}{d_h^\mu(s_h)} \text{Var}_\mu\left[\frac{\pi_h(a_h^{(1)}|s_h)}{\mu_h(a_h^{(1)}|s_h)} Q_h^\pi(s_h, a_h^{(1)})\right], \qquad (4.2)$$

where $Q_h^\pi(s_h, a_h) := \mathbb{E}\left[(V_{h+1}^\pi(s_{h+1}^{(1)}) + r_h^{(1)})\big|s_h^{(1)} = s_h, a_h^{(1)} = a_h\right]$ is the standard $Q$-function the MDP. The gap is significant as the CR lower bound (4.1) itself only has a worst-case bound of $H^2 \tau_s \tau_a / n$ [2], while (4.2) is proportional to $H^3 \tau_s \tau_a / n$. This implies that our estimator is optimal up to a factor of $H$. See Remark 4 for more details in the appendix.

It is an intriguing open question whether this additional factor of $H$ can be removed. Our conjecture is that the answer is negative and what we established in Theorem 4.1 matches the *correct* information-theoretic limit for any methods in the cases when the action space $\mathcal{A}$ is continuous (or significantly larger than $n$). This conjecture is consistent with an existing lower bound in the simpler contextual bandits setting, where Wang et al. [2017] established that a variance of expectation term analogous to the one above cannot be removed, and no estimators can asymptotically attain the CR lower bound for all problems in the large state/action space setting.

## 4.1 Proof Sketch

In this section, we briefly describe the main technical components in the proof of Theorem 4.1. More detailed arguments are deferred to the full proof in Appendix B.

Recall that (3.1) is equivalent to $\sum_{t=1}^{H} \sum_{s_t} \widehat{d_t^\pi}(s_t) \widehat{r}^\pi(s_t)$, where $\widehat{r}^\pi(s_t)$ is estimated with importance sampling and $\widehat{d_t^\pi}(s_t)$ is recursively estimated using $\widehat{d_{t-1}^\pi}(s_{t-1})$ and the importance sampling estimator of the transition matrix $P_t^\pi(s_t|s_{t-1})$ under $\pi$. While the MIS estimator is easy to state, it is not straightforward to analyze. We highlight three challenges below.

1. *Dependent data and complex estimator:* While the episodes are independent, the data within each episode are not. Each time step of our MIS estimator uses the data from all episodes up to that time step.

2. *An annoying bias:* There is a non-zero probability that some states $s_t$ at time $t$ is not visited at all in all $n$ episodes. This creates a bias in the estimator of $\hat{d}_h^\pi$ for all time $h > t$. While the probability of this happening is extremely small, conditioning on the high probability event breaks the natural conditional independences, which makes it hard to analyze.

3. *Error propagation:* The recursive estimator $\hat{d}_t^\pi$ is affected by all estimation errors in earlier time steps. Naive calculation of the error with a constant slack in each step can lead to a "snowball" effect that causes an exponential blow-up.

All these issues require delicate handling because otherwise the MSE calculation will not be tight. Our solutions are as follows.

**Defining the appropriate filtration.** The first observation is that we need to have a convenient representation of the data. Instead of considering the $n$ episodes as independent trajectories, it is more useful to think of them all together as a Markov chain of multi-dimensional observations

of $n$ *state, action, reward* triplets. Specifically, we define the "cumulative" data up to time $t$ by $\text{Data}_t := \left\{ s_{1:t}^{(i)}, a_{1:t-1}^{(i)}, r_{1:t-1}^{(i)} \right\}_{i=1}^{n}$. Also, we observe that the state of the Markov chain at time $t$ can be summarized by $n_{s_t}$ — the number of times state $s_t$ is visited.

**Fictitious estimator technique.** We address the bias issue by defining a fictitious estimator $\tilde{v}^{\pi}$. The fictitious estimator is constructed by, instead of $\hat{d}_t^{\pi}$ and $\hat{r}_t^{\pi}$, the fictitious version of these estimators $\tilde{d}_t^{\pi}$ and $\tilde{r}_t^{\pi}$, where $\tilde{d}_t^{\pi}$ is constructed recursively using

$$\tilde{d}_t^{\pi}(s_t) = \sum_{s_{t-1}} \tilde{P}^{\pi}(s_t|s_{t-1}) \tilde{d}_{t-1}^{\pi}(s_{t-1}).$$

The key difference is that whenever $n_{s_t} < \mathbb{E}_{\mu} n_{s_t}(1-\delta)$ for some $0 < \delta < 1$, we assign $\tilde{P}^{\pi}(s_{t+1}|s_t) = P^{\pi}(s_{t+1}|s_t)$ and $\tilde{r}^{\pi}(s_t) = \mathbb{E}_{\pi}[r_t|s_t]$ — the true values of interest. This ensures that the fictitious estimator is always unbiased (see Lemma B.2). Note that this fictitious estimator cannot be implemented in practice. It is used as a purely theoretical construct that simplifies the analysis of the (biased) MIS estimator. In Lemma B.1, we show that the $\tilde{v}^{\pi}$ and $\hat{v}^{\pi}$ are exponentially close to each other.

**Disentangling the dependency by backwards peeling.** The fictitious estimator technique reduces the problem of calculating the MSE of the MIS estimator to a variance analysis of the fictitious estimator. By recursively applying the law of total variance backwards that peels one item at a time from $\text{Data}_t$, we establish an exact linear decomposition of the variance of the fictitious estimator (Lemma B.3):

$$\text{Var}[\tilde{v}^{\pi}] = \sum_{h=0}^{H} \sum_{s_h} \mathbb{E}\left[ \frac{\widetilde{d}_h^{\pi}(s_h)^2}{n_{s_h}} \mathbf{1}\left( n_{s_h} \geq \frac{nd_h^{\mu}(s_h)}{(1-\delta)^{-1}} \right) \right] \text{Var}_{\mu}\left[ \frac{\pi(a_h^{(1)}|s_h)}{\mu(a_h^{(1)}|s_h)} (V_{h+1}^{\pi}(s_{h+1}^{(1)}) + r_h^{(1)}) \middle| s_h^{(1)} = s_h \right].$$

Observe that the value function $V_t^{\pi}$ shows up naturally. This novel decomposition can be thought of as a generalization of the celebrated Bellman-equation of variance [Sobel, 1982] in the off-policy, episodic MDP setting with a finite sample and can be of independent interest.

**Characterizing the error propagation in $\tilde{d}_h^{\pi}(s_h)$.** Lastly, we bound the error term in the state distribution estimation as follows

$$\mathbb{E}\left[ \frac{\widetilde{d}_h^{\pi}(s_h)^2}{n_{s_h}} \mathbf{1}\left( n_{s_h} \geq \frac{nd_h^{\mu}(s_h)}{(1-\delta)^{-1}} \right) \right] \leq \frac{(1-\delta)^{-1}}{n} \left( \frac{d_h^{\pi}(s_h)^2}{d_h^{\mu}(s_h)} + \text{Var}\left[ \widetilde{d}_h^{\pi}(s_h) \right] \right),$$

which reduces the problem to bounding $\text{Var}[\widetilde{d}_h^{\pi}(s_h)]$. We show (in Theorem B.1) that instead of an exponential blow-up as will a concentration-inequality based argument imply, the variance increases at most linearly in $h$: $\text{Var}[\widetilde{d}_h^{\pi}(s_h)] \leq \frac{2(1-\delta)^{-1}hd_h^{\pi}(s_h)}{n}$. The proof uses a novel decomposition of $\text{Cov}(\widetilde{d}_h^{\pi})$ (Lemma B.5), which is derived using a similar backwards peeling argument as before. Finally, Theorem 4.1 is established by appropriately choosing $\delta = O(\sqrt{\log n/n \min_{t,s_t} d_t^{\mu}(s_t)})$.

Due to space limits, we can only highlight a few key elements of the proof. We invite the readers to check out a more detailed exposition in Appendix B.

## 5 Experiments

Throughout this section, we present the empirical results to illustrate the comparison among different estimators. We demonstrate the effectiveness of our proposed marginalized estimator by comparing it with different classic estimators on several domains.

The methods we compare in this section are: *direct method* (DM), *importance sampling* (IS), *weighted importance sampling* (WIS), *importance sampling with stationary state distribution* (SSD-IS), and *marginalized importance sampling* (MIS). DM uses the model-based approach to estimate $T_t(s_t|s_{t-1}, a_{t-1}), r_t(s_t, a_t)$ by enumerating all tuples of $(s_{t-1}, a_{t-1}, s_t)$, IS is the step-wise importance sampling method, WIS uses the step-wise weighted (self-normalized) importance sampling method, SSD-IS denotes the method of importance sampling with stationary state distribution proposed by [Liu et al., 2018a][3], and MIS is our proposed marginalized method. Note that our MIS

also uses the trick of self-normalization to obtain better performance, but the MIS normalization is different: we normalize the estimate $\widehat{d}_t^\pi$ to the probability simplex, whereas WIS normalizes the importance weights. We provide further results by comparing doubly robust estimator, weighted doubly robust estimator, and our estimators in Appendix D. We use logarithmic scales in all figures and include 95% confidence intervals as error bars from 128 runs. Our metric is the relative root mean squared error (**Relative-RMSE**), which is **the ratio of RMSE and the true value** $v^\pi$.

**Time-invariant MDPs** We first test our methods on the standard ModelWin and ModelFail models with time-invariant MDPs, first introduced by Thomas and Brunskill [2016]. The **ModelWin** domain simulates a fully observable MDP, depicted in Figure 1(a). On the other hand, the **ModelFail** domain (Figure 1(b)) simulates a partially observable MDP, where the agent can only tell the difference between $s_1$ and the "other" unobservable states. A detailed description of these two domains can be found in Appendix D. For both problems, the target policy $\pi$ is to

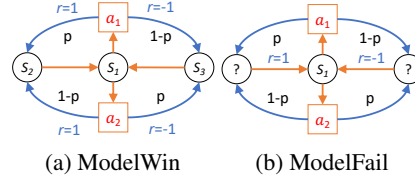

(a) ModelWin        (b) ModelFail

Figure 1: MDPs of OPE domains.

always select $a_1$ and $a_2$ with probabilities 0.2 and 0.8, respectively, and the behavior policy $\mu$ is a uniform policy.

We provide two types of experiments to show the properties of our marginalized approach. The first kind is with different numbers of episodes, where we use a fixed horizon $H = 50$. The second kind is with different horizons, where we use a fixed number of episodes $n = 1024$. We use MIS only with observable states and the partial trajectories between them. Details about applying MIS with partial observability can be found in Appendix C. While this approach is general in more complex applications, for ModelFail, the agent always visits $s_1$ at every other step and we can simply replace $\frac{\pi(a_t^{(i)}|s_t^{(i)})}{\mu(a_t^{(i)}|s_t^{(i)})}$ with $\frac{\pi(a_{2\tau}^{(i)}|s_{2\tau}^{(i)}=?)}{\mu(a_{2\tau}^{(i)}|s_{2\tau}^{(i)}=?)} \frac{\pi(a_{2\tau-1}^{(i)}|s_{2\tau-1}^{(i)})}{\mu(a_{2\tau-1}^{(i)}|s_{2\tau-1}^{(i)})}$ for $t = 2\tau - 1$ in (3.2).

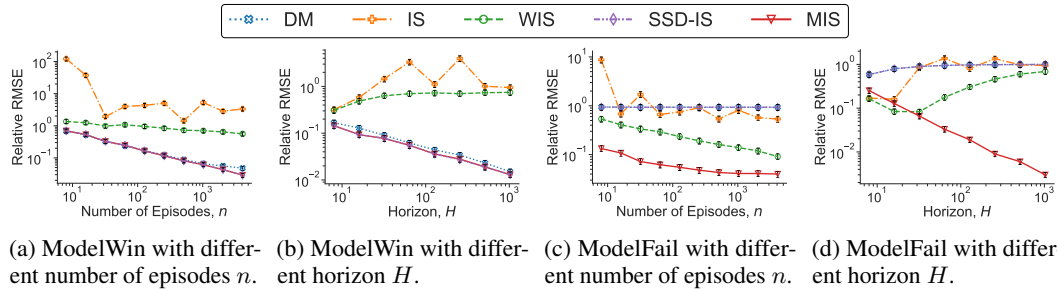

(a) ModelWin with different number of episodes $n$. (b) ModelWin with different horizon $H$. (c) ModelFail with different number of episodes $n$. (d) ModelFail with different horizon $H$.

Figure 2: Results on Time-invariant MDPs. MIS matches DM on ModelWin and outperforms IS/WIS on ModelFail, both of which are the best existing methods on their respective domains.

Figure 2 shows the results in the time-invariant ModelWin MDP and ModelFail MDP. The results clearly demonstrate that MIS maintains a polynomial dependence on $H$ and matches the best alternatives such as DM in Figure 2(b) and IS at the beginning of Figure 2(d). Notably, the IS in Figure 2(d) reflects a bias-variance trade-off, that its RMSE is smaller at short horizons due to unbiasedness yet larger at long horizons due to high variance.

**Time-varying, non-mixing MDPs with continuous actions.** We also test our approach in simulated MDP environments where the states are binary, the actions are continuous between [0,1] and the state transition models are time-varying with a finite horizon $H$. The agent starts at State 1. At every step, the environment samples a random parameter $p \in [0.5/H, 0.5 - 0.5/H]$. Any agent in State 1 will transition to State 0 if and only if it samples an action between $[p - 0.5/H, p + 0.5/H]$. On the other hand, State 0 is a sinking state. The agent collects rewards at State 0 in the latter half of the steps ($t \geq H/2$). Thus, the agent wants to transition to State 0, but the transition probability is inversely proportional to the horizon $H$ for uniform action policies. We pick the behavior policy to be uniform on $[0, 1]$ and the target policy to be uniform on $[0, 0.5]$ with 95% total probability and 5% chance uniformly distributed on $[0.5, 1]$.

Figure 3(a) shows the asymptotic convergence rates of RMSE with respect to the number of episodes, given fixed horizon $H = 64$. MIS converges at a $O(1/\sqrt{n})$ rate from the very beginning. In comparison, neither IS or MIS has entered their asymptotic $n^{-1/2}$ regime yet with $n \leq 4,096$. SSD-IS does not improve as $n$ gets larger, because the stationary state distribution (a point mass on State 0) is not a good approximation of the average probability of visiting State 0 for $t \in [H/2, H]$. We exclude DM because it requires additional model assumptions to apply to continuous action spaces.

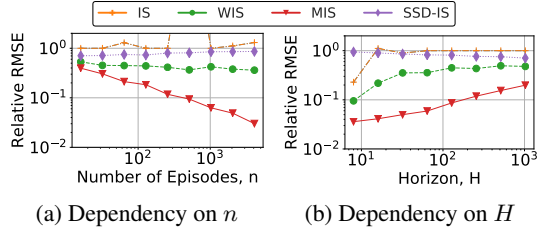

(a) Dependency on $n$      (b) Dependency on $H$

Figure 3: Time-varying MDPs

Figure 3(b) shows the Relative RMSE dependency in $H$, fixing the number of episodes $n = 1024$. We see that as $H$ gets larger, the Relative RMSE scales as $O(\sqrt{H})$ for MIS and stays roughly constant for SSD-IS. Since the true reward $v^\pi \propto H$, the result matches the worst-case bound of a $O(H^3)$ MSE in Corollary 1. SSD-IS saves a factor of $H$ in variance, as it marginalizes over the $H$ steps, but introduces a large bias as we have seen in Figure 3(a). IS and WIS worked better for small $H$, but quickly deteriorates as $H$ increases. Together with Figure 3(a), we may conclude that In conclusion, MIS is the only method, among the alternatives in this example, that produces a consistent estimator with low variance.

**Mountain Car.** Finally, we benchmark our estimator on the Mountain Car domain [Singh and Sutton, 1996], where an under-powered car drives up a steep valley by "swinging" on both sides to gradually build up potential energy. To construct the stochastic behavior policy $\mu$ and stochastic evaluated policy $\pi$, we first compute the optimal Q-function using Q-learning and use its softmax policy of the optimal Q-function as evaluated policy $\pi$ (with the temperature of 1). For the behavior policy $\mu$, we also use the softmax policy of the optimal Q-function but set the temperature to 1.25. Note that this is a finite-horizon MDP with continuous state. We apply MIS by discretizing the state space as in [Jiang and Li, 2016].

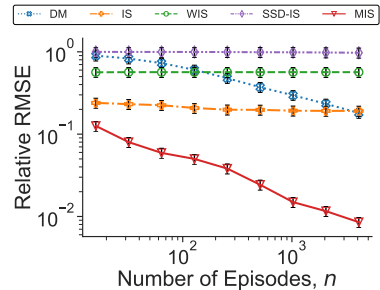

Figure 4: Mountain Car with different number of episodes.

The results, shown in Figure 4, demonstrate the effectiveness of our approach in a common benchmark control task, where the ability to evaluate under long horizons is required for success. Note that Mountain Car is an episodic environment with a absorbing state, so it is not a setting that SSD-IS is designed for. We include the the detailed description on the experimental setup and discussion on the results in Appendix D.

## 6 Conclusions

In this paper, we propose a marginalized importance sampling (MIS) method for the problem of off-policy evaluation in reinforcement learning. Our approach gets rid of the burden of horizon by using an estimated marginal state distribution of the target policy at every step instead of the cumulative product of importance weights.

Comparing to the pioneering work of Liu et al. [2018a] that uses a similar philosophy, this paper focuses on the finite state episodic setting with an potentially infinite action space. We proved the first finite sample error bound for such estimators with polynomial dependence in all parameters. The error bound is tight in that it matches the asymptotic variance of a fictitious estimator that has access to oracle information up to a low-order additive factor. Moreover, it is within a factor of $O(H)$ of the Cramer-Rao lower bound of this problem in [Jiang and Li, 2016]. We conjecture that this additional factor of $H$ is required for any estimators in the *infinite action* setting.

Our experiments demonstrate that the MIS estimator is effective in practice as it achieves substantially better performance than existing approaches in a number of benchmarks.

## Acknowledgement

The authors thank Yu Bai, Murali Narayanaswamy, Lin F. Yang, Nan Jiang, Phil Thomas, Ying Yang for helpful discussion and Amazon internal review committee for the feedback on an early version of the paper. We also acknowledge the NeurIPS area chair, anonymous reviewers for helpful comments and Ming Yin for carefully proofreading the paper.

YW was supported by a start-up grant from UCSB CS department, NSF-OAC 1934641 and a gift from AWS ML Research Award.

## Footnotes

[1]The clipping step to $[0, HR_{\max}]$ or $[0, V_{\max}]$ should not be alarming. It is required only for technical reasons, and the clipped estimator is a valid estimator to begin with. Since the true policy value must be within the range, the clipping step is only going to improve the MSE.

[2] This is somewhat surprising as each of the $H$ summands in the expression can be as large as $H^2$.

[3] Our implementation of SSD-IS for the discrete state case is described in Appendix D.3.

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
