[Supplementary Material]

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

[4]It depends on unknown information such as $d_t^\mu, \mathbb{P}_{t,t-1}^\pi$, exact conditional expectation of the reward $r_t^\pi$ and so on.

[5]These are really not in more precise calculations but are assumed to simplify the statement of our results.

[6]The transition matrices on both $\pi$ and $\mu$ are actually stationary.

[7]In the released code provided by the authors of [Liu et al., 2018a], there is a version of SSD-IS implemented for the discrete state space that first estimates $d_\infty^\pi(s)$ than output the importance weights to be the ratio of this estimate and $\hat{d}_{1:H-1}^\mu$ (see `https://github.com/zt95/infinite-horizon-off-policy-estimation/blob/master/taxi/Density_Ratio_discrete.py`). However, $\hat{d}_\infty^\pi(s)$ is slightly different from the spectral algorithm that we described and it provides a mysterious result that is inconsistent with the stationary distribution that we derived analytically by hands in the example we considered in Figure 3 ($d_\infty^\pi(s = 1) = 1$ with large probability, while the estimated value by running that piece of code is far off).

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

# Appendix

## A    Concentration inequalities

**Lemma A.1** (Multiplicative Chernoff bound [Chernoff et al., 1952] ). *Let $X$ be a Binomial random variable with parameter $p, n$. For any $\delta > 0$, we have that*

$$\mathbb{P}[X > (1+\delta)pn] < \left(\frac{e^\delta}{(1+\delta)^{1+\delta}}\right)^{np}$$

*and*

$$\mathbb{P}[X < (1-\delta)pn] < \left(\frac{e^{-\delta}}{(1-\delta)^{1-\delta}}\right)^{np}.$$

A slightly weaker bound that suffices for our propose is the following:

$$\mathbb{P}[X < (1-\delta)pn] < e^{-\frac{\delta^2 pn}{2}}$$

If we take $\delta = \sqrt{\frac{20 \log(n)}{pn}}$,

$$\mathbb{P}[X < (1-\delta)pn] < n^{-10}.$$

## B    Theoretical analysis of the marginalized IS estimator

Recall that the marginalized IS estimators are of the following form:

$$\widehat{v}^\pi = \sum_{t=1}^H \sum_{s_t} \widehat{d}_t^\pi(s_t)\widehat{r}_t^\pi(s_t),$$

where we recursively estimate the state-marginal under the target policy $\pi$ using

$$\widehat{d}_t^\pi(s_t) = \sum_{s_{t-1}} \widehat{P}_{t-1,t}^\pi(s_t|s_{t-1})\widehat{d}_{t-1}^\pi(s_{t-1}).$$

We focus on the setting where the number of actions is large and possibly unbounded, in which case, we use importance sampling based estimators of $\widehat{P}_{t-1,t}^\pi$ and $\widehat{r}_t^\pi(s_t)$ instead to get bounds that are independent to $A$. Specifically, we use:

$$\widehat{P}_{t-1}^\pi(s_t|s_{t-1}) = \frac{1}{n_{s_{t-1}}} \sum_{i=1}^n \frac{\pi(a_{t-1}^{(i)}|s_{t-1})}{\mu(a_{t-1}^{(i)}|s_{t-1})} \mathbf{1}(s_{t-1}^{(i)} = s_{t-1}, a_{t-1}^{(i)}, s_t^{(i)} = s_t).$$

and

$$\widehat{r}_t^\pi(s_t) = \frac{1}{n_{s_t}} \sum_{i=1}^n \frac{\pi(a_t^{(i)}|s_t)}{\mu(a_t^{(i)}|s_t)} r_t^{(i)} \mathbf{1}(s_t^{(i)} = s_t).$$

The main challenge in analyzing these involves finding a way to decompose the error in the face of the complex recursive structure, as well as to deal with the bias of the estimator.

**Constructing a fictitious estimator.**    Our proof makes novel use of a fictitious estimator $\widetilde{v}^\pi$ which uses $\widetilde{d}_t^\pi = \widehat{P}_{t+1,t}^\pi \widetilde{d}_{t-1}^\pi$ and $\widetilde{r}_t^\pi$ instead of $\widehat{d}_t^\pi = \widehat{P}_{t+1,t}^\pi(\cdot|s_t)\widehat{d}_{t-1}^\pi$ and $\widehat{r}_t^\pi$ in the original estimator $\widehat{v}^\pi$.

To write it down more formally,

$$\widetilde{v}^\pi := \sum_{t=1}^H \sum_{s_t} \widetilde{d}_t^\pi(s_t)\widetilde{r}_t^\pi(s_t)$$

where $\widetilde{d}_t^\pi(s_t)$ is constructed recursively using

$$\widetilde{d}_t^\pi = \widetilde{\mathbb{P}}_{t,t-1}^\pi \widetilde{d}_{t-1}^\pi$$

as in our regular estimator for $t = 2, 3, 4, ..., H$, and $\widetilde{d}_1^\pi = \widehat{d}_1$. In particular,

$$\widetilde{r}_t^\pi(s_t) = \begin{cases} \widehat{r}_t^\pi(s_t) & \text{if } n_{s_t} \geq nd_t^\mu(s_t)(1-\delta) \\ r_t^\pi(s_t) & \text{otherwise;} \end{cases}$$

and

$$\widetilde{\mathbb{P}}_{t,t-1}^\pi(\cdot|s_{t-1}) = \begin{cases} \widehat{\mathbb{P}}_{t,t-1}^\pi & \text{if } n_{s_{t-1}} \geq nd_t^\mu(s_{t-1})(1-\delta) \\ \mathbb{P}_{t,t-1}^\pi & \text{otherwise.} \end{cases}$$

In the above, $0 < \delta < 1$ is a parameter that we will choose later.

This estimator $\widetilde{v}^\pi$ is fictitious because it is *not implementable* using the data[4], but it is somewhat easier to work with and behaves essentially the same as our actual estimator $\widehat{v}^\pi$. As a result, we can analyze our estimator through analyzing $\widetilde{v}^\pi$. The following lemma formalizes the idea.

**Lemma B.1.** *Let $\widehat{v}^\pi$ be our MIS estimator and $\mathcal{P}$ be the projection operator to $[0, HR_{\max}]$ and $\widetilde{v}^\pi$ be the unbiased fictitious estimator that we described above with parameter $\delta$. The MSE of the clipped version of our MIS estimator obeys*

$$\mathbb{E}[(\mathcal{P}\widehat{v}^\pi - v^\pi)^2] \leq \mathbb{E}[(\widetilde{v}^\pi - v^\pi)^2] + 3H^3SR_{\max}^2 e^{-\frac{\delta^2 n \min_{t,s_t} d_t^\mu(s_t)}{2}}$$

*Proof of Lemma B.1.* Let $E$ denotes the event of $\{\exists t, s_t, \text{ s.t. } n_{s_t} < nd_t^\mu(s_t)(1-\delta)\}$. Let $\mathcal{P}_E$ be the *conditional* projection operator that clips the value to $[0, HR_{\max}]$ whenever $E$ is true. Note that for any $x \in \mathbb{R}$, we have $\mathcal{P}(\mathcal{P}_E x) = \mathcal{P}x$. By the non-expansiveness of $\mathcal{P}$,

$$\mathbb{E}[(\mathcal{P}\widehat{v}^\pi - v^\pi)^2] \leq \mathbb{E}[(\mathcal{P}_E\widehat{v}^\pi - v^\pi)^2] = \mathbb{E}[(\mathcal{P}_E\widehat{v}^\pi - \mathcal{P}_E\widetilde{v}^\pi + \mathcal{P}_E\widetilde{v}^\pi - v^\pi)^2]$$

$$= \mathbb{E}[(\mathcal{P}_E\widehat{v}^\pi - \mathcal{P}_E\widetilde{v}^\pi)^2] + 2\mathbb{E}[(\mathcal{P}_E\widehat{v}^\pi - \mathcal{P}_E\widetilde{v}^\pi)(\mathcal{P}_E\widetilde{v}^\pi - v^\pi)] + \mathbb{E}[(\mathcal{P}_E\widetilde{v}^\pi - v^\pi)^2]$$

$$= \mathbb{P}[E]\mathbb{E}\left[(\mathcal{P}_E\widehat{v}^\pi - \mathcal{P}_E\widetilde{v}^\pi)^2 + 2(\mathcal{P}_E\widehat{v}^\pi - \mathcal{P}_E\widetilde{v}^\pi)(\mathcal{P}_E\widetilde{v}^\pi - v^\pi)\big|E\right] + \mathbb{P}[E^c] \cdot 0 + \mathbb{E}[(\mathcal{P}_E\widetilde{v}^\pi - \mathcal{P}_E v^\pi)^2]$$

$$\leq 3\mathbb{P}[E]H^2R_{\max}^2 + \mathbb{E}[(\widetilde{v}^\pi - v^\pi)^2].$$

The third line is by the law of total expectation and the fact that whenever $E$ is not true, $\widehat{v}^\pi = \widetilde{v}^\pi$. The last line uses the fact that $\mathcal{P}_E\widehat{v}^\pi, \mathcal{P}_E\widetilde{v}^\pi, v^\pi$ are all within $[0, HR_{\max}]$ when conditioning on $E$ as well as the non-expansiveness of the projection operator which implies that

$$\mathbb{E}[(\mathcal{P}_E(\widetilde{v}^\pi - v^\pi))^2] \leq \mathbb{E}[(\widetilde{v}^\pi - v^\pi)^2].$$

It remains to bound $\mathbb{P}[E]$. By the multiplicative Chernoff bound (Lemma A.1 in the Appendix) we get that

$$\mathbb{P}\left[n_{s_t} < nd_t^\mu(s_t)(1-\delta)\right] \leq e^{-\frac{\delta^2 nd_t^\mu(s_t)}{2}}$$

By a union bound over each $t$ and $s_t$, we have

$$\mathbb{P}[E] \leq \sum_t \sum_{s_t} \mathbb{P}[n_{s_t,t} < nd_t^\mu(s_t)(1-\delta)] \leq HSe^{-\frac{\delta^2 n \min_{t,s_t} d_t^\mu(s_t)}{2}}$$

as stated. $\qquad\square$

Lemma B.1 establishes that when $n \geq \frac{\text{polylog}(S,H,n)}{\min_{t,s_t} d_t^\mu(s_t)}$, we can bound the MSE of a projected version of our estimator using the MSE of the fictitious estimator. The projection to $[0, HR_{\max}]$ is a post-processing that we needed in our proof for technical reasons, and we know that $\mathbb{E}[(\mathcal{P}\widehat{v}^\pi - v^\pi)^2] \leq \mathbb{E}[(\widehat{v}^\pi - v^\pi)^2]$ so it only improves the performance.

**Properties of the Fictitious Estimator.** Now let us prove that $\widetilde{v}^\pi$ is unbiased and also analyze its variance. Recall that the estimator is the following:

$$\widetilde{v}^\pi = \sum_{t=1}^H \sum_{s_t} \widetilde{d}_t^\pi(s_t)\widetilde{r}_t^\pi(s_t) = \sum_{t=1}^H \langle \widetilde{d}_t^\pi, \widetilde{r}_t^\pi \rangle$$

where we denote quantities $\widetilde{d}_t^\pi, \widetilde{r}_t^\pi$ in vector forms in $\mathbb{R}^S$.

In the remainder of this section, we will use $E_t$ as a short hand to denote the event such that $\{n_{s_t} \geq nd_t^\mu(s_t)(1-\delta)\}$, and $\mathbf{1}(E_t)$ be the corresponding indicator function.

**Lemma B.2** (Unbiasedness of $\widetilde{v}^\pi$). $\mathbb{E}[\widetilde{v}^\pi] = v^\pi$ *for all* $\delta < 1$.

*Proof of Lemma B.2.* The idea of the proof is to recursively apply the Law of Total Expectation backwards from the last round by taking conditional expectations. For simplicity of the proof we will denote

$$\text{Data}_t := \left\{ s_{1:t}^{(i)}, a_{1:t-1}^{(i)}, r_{1:t-1}^{(i)} \right\}_{i=1}^n.$$

Also, in the base case, let's denote $\text{Data}_1 := \left\{ s_1^{(i)} \right\}_{i=1}^n$ and that $r_t^\pi(s_t) := \mathbb{E}_\pi[r_t^{(1)} | s_t^{(1)} = s_t]$

We first making a few observations that will be useful in the arguments that follow. Firstly, $\widetilde{d}_t^\pi$ and $\widetilde{r}_{t-1}^\pi$ are deterministic given $\text{Data}_t$. Secondly,

$$\mathbb{E}[\widetilde{P}_{t,t-1}^\pi | \text{Data}_{t-1}] = P_{t,t-1}^\pi, \quad \text{and} \quad \mathbb{E}[\widetilde{r}_t^\pi | \text{Data}_t] = r_t^\pi.$$

These observations are true for all $t = 1, ..., H$. To see the unbiasedness of the conditional expectation, note that when $n_{s_t} \geq n d_t^\mu(s_t)(1-\delta)$, the estimators are just empirical mean, which are unbiased and when $n_{s_t} < n d_t^\mu(s_t)(1-\delta)$, we also have an unbiased estimator by the construction of the fictitious estimator. For all $\delta < 1$, the case $n_{s_t} = 0$ is ruled out. Thirdly, we write down the standard Bellman equation for policy $\pi$

$$V_h(s_h) = r_h^\pi(s_h) + \sum_{s_{h+1}} P_{h+1,h}^\pi(s_{h+1}|s_h) V_{h+1}(s_{h+1}).$$

where $V_h(s_h) := \mathbb{E}_\pi \left[ \sum_{t=h}^H r_t^{(1)} \middle| s_t^{(1)} = s_h \right]$ or in a matrix form

$$V_h = r_h^\pi + [P_{h+1,h}^\pi]^T V_{h+1}.$$

These observations together allow us to write the following recursion:

$$\mathbb{E}\left[ \langle \widetilde{d}_h^\pi, V_h^\pi \rangle + \sum_{t=1}^{h-1} \langle \widetilde{d}_t^\pi, \widetilde{r}_t^\pi \rangle \middle| \text{Data}_{h-1} \right]$$

$$= \langle \mathbb{E}[\widetilde{P}_{h,h-1}^\pi | \text{Data}_{h-1}] \widetilde{d}_{h-1}^\pi, V_h^\pi \rangle + \langle \widetilde{d}_{h-1}^\pi, \mathbb{E}[\widetilde{r}_{h-1}^\pi | \text{Data}_{h-1}] \rangle + \sum_{t=1}^{h-2} \langle \widetilde{d}_t^\pi, \widetilde{r}_t^\pi \rangle$$

$$= \langle \widetilde{d}_{h-1}^\pi, [P_{h,h-1}^\pi]^T V_h^\pi + r_{h-1}^\pi \rangle + \sum_{t=1}^{h-2} \langle \widetilde{d}_t^\pi, \widetilde{r}_t^\pi \rangle$$

$$\underset{\substack{\uparrow \\ \text{Bellman equation}}}{=} \langle \widetilde{d}_{h-1}^\pi, V_{h-1}^\pi \rangle + \sum_{t=1}^{h-2} \langle \widetilde{d}_t^\pi, \widetilde{r}_t^\pi \rangle.$$

Finally, by taking (full) expectation and chaining the above recursions together, we get

$$\mathbb{E}\left[ \sum_{t=1}^H \langle \widetilde{d}_t^\pi, \widetilde{r}_t^\pi \rangle \right] = \mathbb{E}\left[ \langle \widetilde{d}_H^\pi, V_H^\pi \rangle + \sum_{t=1}^{H-1} \langle \widetilde{d}_t^\pi, \widetilde{r}_t^\pi \rangle \right]$$

$$= \mathbb{E}\left[ \langle \widetilde{d}_{H-1}^\pi, V_{H-1}^\pi \rangle + \sum_{t=1}^{H-2} \langle \widetilde{d}_t^\pi, \widetilde{r}_t^\pi \rangle \right]$$

$$= \dots$$

$$= \mathbb{E}\left[ \langle \widetilde{d}_1^\pi, V_1^\pi \rangle \right] = v^\pi,$$

which concludes the proof. $\square$

Now let's tackle the variance of the fictitious estimator.

**Lemma B.3** (Variance decomposition).

$$\text{Var}[\widetilde{v}^\pi] = \frac{\text{Var}[V_1^\pi(s_1^{(1)})]}{n}$$

$$+ \sum_{h=1}^{H} \sum_{s_h} \mathbb{E}\left[\frac{\widetilde{d}_h^\pi(s_h)^2}{n_{s_h}}\mathbf{1}(E_h)\right] \text{Var}_\mu\left[\frac{\pi(a_h^{(1)}|s_h)}{\mu(a_h^{(1)}|s_h)}(V_{h+1}^\pi(s_{h+1}^{(1)}) + r_h^{(1)})\bigg|s_h^{(1)} = s_h\right].$$

where $V_t^\pi(s_t)$ denotes the value function under $\pi$ which satisfies the Bellman equation

$$V_t^\pi(s_t) = r_t^\pi(s_t) + \sum_{s_{t+1}} P_t^\pi(s_{t+1}|s_t)V_{t+1}^\pi(s_{t+1}).$$

**Remark 1.** *The decomposition of variance is very interpretable. The first part of the variance is coming from estimating the initial state. The second part is coming from the conditional variance of estimating $P_{t+1,t}^\pi(s_t)$ and $r_t^\pi(s_t)$ using importance sampling over $a_t$.*

*Proof of Lemma B.3.* The proof uses a peeling argument that recursively applies the law of total variance from the last time point backwards.

The key of the argument relies upon the following identity that holds for all $h = 1, ..., H - 1$.

$$\text{Var}\left[\langle\widetilde{d}_{h+1}^\pi, V_{h+1}^\pi\rangle + \sum_{t=1}^{h}\langle\widetilde{d}_t^\pi, \widetilde{r}_t^\pi\rangle\right] = \mathbb{E}\left[\text{Var}\left[\langle\widetilde{d}_{h+1}^\pi, V_{h+1}^\pi\rangle + \langle\widetilde{d}_h^\pi, \widetilde{r}_h^\pi\rangle\Big|\text{Data}_h\right]\right]$$

$$+ \text{Var}\left[\langle\widetilde{d}_h^\pi, V_h^\pi\rangle + \sum_{t=1}^{h-1}\langle\widetilde{d}_t^\pi, \widetilde{r}_t^\pi\rangle\right]. \tag{B.1}$$

Note that in (B.1), when we condition on $\text{Data}_h$, $\widetilde{d}_h^\pi$ is fixed. Also, $\widetilde{P}_{h+1,h}(\cdot, s_h)$ and $\widetilde{r}_h^\pi(s_h)$ for each $s_h$ are conditionally independent given $\text{Data}_h$, since $\text{Data}_h$ partitions the $n$ episodes into $S$ disjoint sets according to the states $s_h^{(i)}$ at time $h$. These observations imply that

$$\mathbb{E}\left[\text{Var}\left[\langle\widetilde{d}_{h+1}^\pi, V_{h+1}^\pi\rangle + \langle\widetilde{d}_h^\pi, \widetilde{r}_h^\pi\rangle\Big|\text{Data}_h\right]\right]$$

$$= \mathbb{E}\left[\sum_{s_h}\text{Var}\left[\widetilde{d}_h^\pi(s_h)\langle\widetilde{P}_{h+1,h}(\cdot, s_h), V_{h+1}^\pi\rangle + \widetilde{d}_h^\pi(s_h)\cdot\widetilde{r}_h^\pi(s_h)\Big|\text{Data}_h\right]\right]$$

$$= \mathbb{E}\left[\sum_{s_h}\mathbf{1}(E_h)\text{Var}\left[\widetilde{d}_h^\pi(s_h)\langle\widetilde{P}_{h+1,h}(\cdot, s_h), V_{h+1}^\pi\rangle + \widetilde{d}_h^\pi(s_h)\cdot\widetilde{r}_h^\pi(s_h)\Big|\text{Data}_h\right]\right]$$

$$= \mathbb{E}\left[\sum_{s_h}\mathbf{1}(E_h)\text{Var}\left[\left\langle\frac{\widetilde{d}_h^\pi(s_h)}{n_{s_h}}\sum_{i|s_h^{(i)}=s_h}\frac{\pi(a_h^{(i)}|s_h)}{\mu(a_h^{(i)}|s_h)}\mathbf{e}_{s_{h+1}^{(i)}}, V_{h+1}^\pi\right\rangle + \frac{\widetilde{d}_h^\pi(s_h)}{n_{s_h}}\sum_{i|s_h^{(i)}=s_h}\frac{\pi(a_h^{(i)}|s_h)}{\mu(a_h^{(i)}|s_h)}r_h^{(i)}\bigg|\text{Data}_h\right]\right]$$

$$= \mathbb{E}\left[\sum_{s_h}\widetilde{d}_h^\pi(s_h)^2\mathbf{1}(E_h)\text{Var}\left[\frac{1}{n_{s_h}}\sum_{i|s_h^{(i)}=s_h}\frac{\pi(a_h^{(i)}|s_h)}{\mu(a_h^{(i)}|s_h)}(V_{h+1}^\pi(s_{h+1}^{(i)}) + r_h^{(i)})\bigg|\text{Data}_h\right]\right]$$

$$= \sum_{s_h}\mathbb{E}\left[\frac{\widetilde{d}_h^\pi(s_h)^2}{n_{s_h}}\mathbf{1}(E_h)\right]\text{Var}\left[\frac{\pi(a_h^{(1)}|s_h)}{\mu(a_h^{(1)}|s_h)}(V_{h+1}^\pi(s_{h+1}^{(1)}) + r_h^{(1)})\bigg|s_h^{(1)} = s_h\right]. \tag{B.2}$$

The second line uses the conditional independence we mentioned above. The third line uses that when $n_{s_h} < nd_h^\mu(s_h)$, the conditional variance is 0. The fourth and fifth line apply the definition of the importance sampling estimators and finally the last line uses that the episodes are iid.

Apply (B.1) recursively

$$\mathrm{Var}[\widetilde{v}^\pi] = \mathbb{E}\mathrm{Var}[\widetilde{v}^\pi|\mathrm{Data}_H] + \mathrm{Var}[\mathbb{E}[\widetilde{v}^\pi|\mathrm{Data}_H]]$$

$$= \mathbb{E}\left[\mathrm{Var}[\langle\widetilde{d}_H^\pi, \widetilde{r}_H^\pi\rangle|\mathrm{Data}_H]\right] + \mathrm{Var}[\mathbb{E}[\langle\widetilde{d}_H^\pi, \widetilde{r}_H^\pi\rangle|\mathrm{Data}_H] + \sum_{t=1}^{H-1}\langle\widetilde{d}_t^\pi, \widetilde{r}_t^\pi\rangle]$$

$$= \mathbb{E}\left[\mathrm{Var}[\langle\widetilde{d}_H^\pi, \widetilde{r}_H^\pi\rangle|\mathrm{Data}_H]\right] + \mathrm{Var}[\langle\widetilde{d}_H^\pi, r_H^\pi\rangle + \sum_{t=1}^{H-1}\langle\widetilde{d}_t^\pi, \widetilde{r}_t^\pi\rangle]$$

$$= \mathbb{E}\left[\mathrm{Var}[\langle\widetilde{d}_H^\pi, \widetilde{r}_H^\pi\rangle|\mathrm{Data}_H]\right] + \mathrm{Var}[\langle\widetilde{d}_H^\pi, V_H^\pi\rangle + \sum_{t=1}^{H-1}\langle\widetilde{d}_t^\pi, \widetilde{r}_t^\pi\rangle]$$

$$= \mathbb{E}\left[\mathrm{Var}[\langle\widetilde{d}_H^\pi, \widetilde{r}_H^\pi\rangle|\mathrm{Data}_H]\right] + \mathbb{E}\left[\mathrm{Var}\left[\langle\widetilde{d}_H^\pi, V_H^\pi\rangle + \langle\widetilde{d}_{H-1}^\pi, \widetilde{r}_{H-1}^\pi\rangle\Big|\mathrm{Data}_{H-1}\right]\right]$$

$$\quad + \mathrm{Var}\left[\langle\widetilde{d}_{H-1}^\pi, V_{H-1}^\pi\rangle + \sum_{t=1}^{H-2}\langle\widetilde{d}_t^\pi, \widetilde{r}_t^\pi\rangle\right]$$

$$= \mathbb{E}\left[\mathrm{Var}[\langle\widetilde{d}_H^\pi, \widetilde{r}_H^\pi\rangle|\mathrm{Data}_H]\right] + \sum_{h=H-1}^{H}\mathbb{E}\left[\mathrm{Var}\left[\langle\widetilde{d}_h^\pi, V_h^\pi\rangle + \langle\widetilde{d}_{h-1}^\pi, \widetilde{r}_{h-1}^\pi\rangle\Big|\mathrm{Data}_{h-1}\right]\right]$$

$$\quad + \mathrm{Var}\left[\langle\widetilde{d}_{H-2}^\pi, V_{H-2}^\pi\rangle + \sum_{t=1}^{H-3}\langle\widetilde{d}_t^\pi, \widetilde{r}_t^\pi\rangle\right]$$

$$= \mathbb{E}\left[\mathrm{Var}[\langle\widetilde{d}_H^\pi, \widetilde{r}_H^\pi\rangle|\mathrm{Data}_H]\right] + \sum_{h=2}^{H}\mathbb{E}\left[\mathrm{Var}\left[\langle\widetilde{d}_h^\pi, V_h^\pi\rangle + \langle\widetilde{d}_{h-1}^\pi, \widetilde{r}_{h-1}^\pi\rangle\Big|\mathrm{Data}_{h-1}\right]\right] + \mathrm{Var}\left[\langle\widetilde{d}_1^\pi, V_1^\pi\rangle\right]$$

Use the boundary condition $V_{H+1} \equiv 0$ as stated in the theorem and apply (B.2), we get that

$$\mathrm{Var}[\widetilde{v}^\pi] = \frac{\mathrm{Var}[V_1^\pi(s_1^{(1)})]}{n}$$
$$+ \sum_{h=1}^{H}\sum_{s_h}\mathbb{E}\left[\frac{\widetilde{d}_h^\pi(s_h)^2}{n_{s_h}}\mathbf{1}(E_h)\right]\mathrm{Var}\left[\frac{\pi(a_h^{(1)}|s_h)}{\mu(a_h^{(1)}|s_h)}(V_{h+1}^\pi(s_{h+1}^{(1)}) + r_h^{(1)})\bigg|s_h^{(1)} = s_h\right].$$

This completes the proof. □

**Bounding the importance weights**   It remains to show that for all $h, s_h$,

$$\mathbb{E}\left[\frac{\widetilde{d}_h^\pi(s_h)^2}{n_{s_h}}\mathbf{1}(E_h)\right] \approx \frac{d_h^\pi(s_h)^2}{nd_h^\mu(s_h)}.$$

By the non-negativity of $\widetilde{d}_h^\pi(s_h)^2$

$$\mathbb{E}\left[\frac{\widetilde{d}_h^\pi(s_h)^2}{n_{s_h}}\mathbf{1}(E_h)\right] \le \frac{(1-\delta)^{-1}}{nd_h^\mu(s_h)}\mathbb{E}\left[\widetilde{d}_h^\pi(s_h)^2\right] = \frac{(1-\delta)^{-1}}{nd_h^\mu(s_h)}(d_h^\pi(s_h)^2 + \mathrm{Var}[\widetilde{d}_h^\pi(s_h)]). \quad \text{(B.3)}$$

where the last identity is true because $\widetilde{d}_h^\pi$ is an unbiased estimator of $d_h^\pi(s_h)$ as the following lemma establishes.

**Lemma B.4** (Unbiasedness of $\widetilde{d}_h^\pi$). *For all $h = 1, ..., H$, the fictitious state marginal estimators are unbiased, that is,*

$$\mathbb{E}[\widetilde{d}_h^\pi] = d_h^\pi.$$

*Proof of Lemma B.4.*   Recall the recursive relationship by construction

$$\widetilde{d}_h^\pi = \widetilde{\mathbb{P}}_{h,h-1}^\pi \widetilde{d}_{h-1}^\pi.$$

We will prove by induction on $h$. First, take the base case $h = 1$: $\mathbb{E}[\widetilde{d}_1^\pi] = \mathbb{E}[\widehat{d}_1^\pi] = d_1^\pi$. Now if $\mathbb{E}[\widetilde{d}_{h-1}^\pi] = d_{h-1}^\pi$, then by the law of total expectation:

$$\mathbb{E}[\widetilde{d}_h^\pi] = \mathbb{E}\left[\mathbb{E}[\widetilde{\mathbb{P}}_{h,h-1}^\pi \widetilde{d}_{h-1}^\pi | \text{Data}_{h-1}]\right]$$
$$= \mathbb{P}_{h,h-1}^\pi \mathbb{E}\left[\widetilde{d}_{h-1}^\pi\right] = \mathbb{P}_{h,h-1}^\pi d_{h-1}^\pi = d_h^\pi.$$

This completes the proof for all $h$. $\qquad\qquad\qquad\qquad\qquad\qquad\qquad\qquad\qquad\qquad$ $\square$

So the problem reduces to bounding $\text{Var}[\widetilde{d}_h^\pi(s_h)]$. We will prove something more useful by bounding the covariance matrix of $\widetilde{d}_h^\pi(s_h)$ in semidefinite ordering.

**Lemma B.5** (Covariance of $\widetilde{d}_h^\pi$).

$$\text{Cov}(\widetilde{d}_h^\pi)$$
$$\preceq \frac{(1-\delta)^{-1}}{n} \sum_{t=2}^{h} \mathbb{P}_{h,t}^\pi \text{diag}\left[\sum_{s_{t-1}} \frac{d_{t-1}^\pi(s_{t-1})^2 + \text{Var}(\widetilde{d}_{t-1}^\pi(s_{t-1}))}{d_{t-1}^\mu(s_{t-1})} \sum_{a_{t-1}} \frac{\pi(a_{t-1}|s_{t-1})^2}{\mu(a_{h-1}|s_{t-1})} \mathbb{P}_{t,t-1}(\cdot|s_{t-1},a_{t-1})\right] [\mathbb{P}_{h,t}^\pi]^T$$
$$+ \frac{1}{n} \mathbb{P}_{h,1}^\pi \text{diag}\left[d_1^\pi\right][\mathbb{P}_{h,1}^\pi]^T.$$

*where $\mathbb{P}_{h,t}^\pi = \mathbb{P}_{h,h-1}^\pi \cdot \mathbb{P}_{h-1,h-2}^\pi \cdots \mathbb{P}_{t+1,t}^\pi$ — the transition matrices under policy $\pi$ from time $t$ to $h$ (define $\mathbb{P}_{h,h}^\pi := I$).*

Before proving the result, let us connect it to what we need in (B.3).

**Corollary 2.** *For $h = 1$, we have:*

$$\text{Var}[\widetilde{d}_1^\pi(s_1)] = \frac{1}{n}(d_h^\pi(s_1) - d_h^\pi(s_1)^2).$$

*For $h = 2, 3, ..., H$, we have:*

$$\text{Var}[\widetilde{d}_h^\pi(s_h)] \leq \frac{(1-\delta)^{-1}}{n} \sum_{t=2}^{h} \sum_{s_t} \mathbb{P}_{h,t}^\pi(s_h|s_t)^2 \varrho(s_t) + \frac{1}{n} \sum_{s_1} \mathbb{P}_{h,1}^\pi(s_h|s_1)^2 d_1(s_1)$$

*where $\varrho(s_t) := \sum_{s_{t-1}} \left(\frac{d_{t-1}^\pi(s_{t-1})^2 + \text{Var}(\widetilde{d}_{t-1}^\pi(s_{t-1}))}{d_{t-1}^\mu(s_{t-1})} \sum_{a_{t-1}} \frac{\pi(a_{t-1}|s_{t-1})^2}{\mu(a_{t-1}|s_{t-1})} \mathbb{P}_{t,t-1}(s_t|s_{t-1},a_{t-1})\right).$*

Note that we have $\text{Var}[\widetilde{d}_{t-1}^\pi(s_{t-1})]$ on the RHS of the equation, which suggests that we in fact need to recursively apply our bounds from $h = 1$ to obtain the overall bound.

**Theorem B.1** (Error propagation). *Let $\tau_a := \max_{t,s_t,a_t} \frac{\pi(a_t|s_t)}{\mu(a_t|s_t)}$ and $\tau_s := \max_{t,s_t} \frac{d_t^\pi(s_t)}{d_t^\mu(s_t)}$[5]. If $n \geq \frac{2(1-\delta)^{-1} t \tau_a \tau_s}{\max\{d_t^\pi(s_t), d_t^\mu(s_t)\}}$ for all $t = 2, ..., H$, then for all $h = 1, 2, ..., H$ and $s_h$, we have that:*

$$\text{Var}[\widetilde{d}_h^\pi(s_h)] \leq \frac{2(1-\delta)^{-1} h \tau_a \tau_s}{n} d_h^\pi(s_h).$$

*Proof of Theorem B.1.* We prove by induction. The base case for $h = 1$ is trivially true because

$$\text{Var}[\widetilde{d}_1^\pi(s_1)] = \frac{1}{n}(d_1^\pi(s_1) - d_1^\pi(s_1)^2) \leq \frac{2(1-\delta)^{-1} \tau_a \tau_s}{n} d_1^\pi(s_1).$$

since $\tau_a \geq 1$ and $\tau_s \geq 1$ by construction.

Assume $\text{Var}[\widetilde{d}_t^\pi(s_t)] \leq \frac{2(1-\delta)^{-1} t \tau_a \tau_s}{n} d_t^\pi(s_t)$ is true for all $t = 1, ..., h-1$, then by our assumption on $n$ and that $h \leq H$, we obtain that

$$\text{Var}[\widetilde{d}_t^\pi(s_t)] \leq d_t^\pi(s_t) \max\{d_t^\pi(s_t), d_t^\mu(s_t)\}$$

for all $t = 1, ..., h - 1$. Plug this into Corollary 2, we get that

$$\varrho(s_t) \leq \sum_{s_{t-1}} \left( d_{t-1}^\pi(s_{t-1}) \frac{2 \max\{d_{t-1}^\pi(s_{t-1}), d_{t-1}^\mu(s_{t-1})\}}{d_{t-1}^\mu(s_{t-1})} \sum_{a_{h-1}} \frac{\pi(a_{t-1}|s_{t-1})^2}{\mu(a_{t-1}|s_{t-1})} \mathbb{P}_{t,t-1}(s_t|s_{t-1}, a_{t-1}) \right)$$

$$\leq 2\tau_s\tau_a \sum_{s_{t-1}} d_{t-1}^\pi(s_{t-1}) \sum_{a_{h-1}} \pi(a_{t-1}|s_{t-1})\mathbb{P}_{t,t-1}(s_t|s_{t-1}, a_{t-1})$$

$$= 2\tau_s\tau_a d_t^\pi(s_t),$$

and that

$$\text{Var}[\widetilde{d}_h^\pi(s_h)] \leq \frac{2(1-\delta)^{-1}\tau_s\tau_a}{n} \sum_{t=2}^{h} \sum_{s_t} \mathbb{P}_{h,t}^\pi(s_h|s_t)^2 d_t^\pi(s_t) + \frac{1}{n}\sum_{s_1}\mathbb{P}_{h,1}^\pi(s_h|s_1)^2 d_1(s_1)$$

$$\leq \frac{2(1-\delta)^{-1}\tau_s\tau_a}{n} \sum_{t=1}^{h} \sum_{s_t} \mathbb{P}_{h,t}^\pi(s_h|s_t)^2 d_t^\pi(s_t)$$

$$\leq \frac{2(1-\delta)^{-1}\tau_s\tau_a}{n} \sum_{t=1}^{h} \sum_{s_t} \mathbb{P}_{h,t}^\pi(s_h|s_t) d_t^\pi(s_t)$$

$$= \frac{2(1-\delta)^{-1}h\tau_s\tau_a}{n} d_h^\pi(s_h)$$

The second inequality uses that $\tau_s, \tau_a \geq 1$, the third inequality uses that $0 \leq \mathbb{P}_{h,t}^\pi(s_h|s_t) \leq 1$. $\qquad\square$

Note that the bound is tight and it implies that the error propagation is moderate. Instead of increasing exponentially, the error increases only linearly in time horizon, as long as $n$ is at least linear in $h$.

*Proof of Lemma B.5.* We start by applying the law of total variance to obtain the following recursive equation

$$\text{Cov}[\widetilde{d}_h^\pi] = \mathbb{E}\left[\text{Cov}\left[\widetilde{\mathbb{P}}_{h,h-1}^\pi \widetilde{d}_{h-1}^\pi \middle| \text{Data}_{h-1}\right]\right] + \text{Cov}\left[\mathbb{E}\left[\widetilde{\mathbb{P}}_{h,h-1}^\pi \widetilde{d}_{h-1}^\pi \middle| \text{Data}_{h-1}\right]\right]$$

$$= \mathbb{E}\left[\text{Cov}\left[\sum_{s_{h-1}} \widetilde{\mathbb{P}}_{h,h-1}^\pi(\cdot|s_{h-1})\widetilde{d}_{h-1}^\pi(s_{h-1}) \middle| \text{Data}_{h-1}\right]\right] + \text{Cov}\left[\mathbb{E}\left[\widetilde{\mathbb{P}}_{h,h-1}^\pi \widetilde{d}_{h-1}^\pi \middle| \text{Data}_{h-1}\right]\right]$$

$$= \underbrace{\mathbb{E}\left[\sum_{s_{h-1}} \text{Cov}\left[\widetilde{\mathbb{P}}_{h,h-1}^\pi(\cdot|s_{h-1}) \middle| \text{Data}_{h-1}\right] \widetilde{d}_{h-1}^\pi(s_{h-1})^2\right]}_{(***)} + \mathbb{P}_{h,h-1}^\pi \text{Cov}[\widetilde{d}_{h-1}^\pi][\mathbb{P}_{h,h-1}^\pi]^T.$$

(B.4)

The decomposition of the covariance in the third line uses that $\text{Cov}(X + Y) = \text{Cov}(X) + \text{Cov}(Y)$ when $X$ and $Y$ are statistically independent. Note that $n_{s_{h-1}}, \widetilde{d}_{h-1}^\pi(s_{h-1})$ are fixed and the columns

of $\widetilde{\mathbb{P}}_{h,h-1}$ are independent when conditioning on $\text{Data}_{h-1}$.

$$(***) = \mathbb{E}\left[\sum_{s_{h-1}} \text{Cov}\left[\frac{1}{n_{s_{h-1}}}\sum_{i=1}^{n} \frac{\pi(a_{h-1}^{(i)}|s_{h-1}^{(i)})}{\mu(a_{h-1}^{(i)}|s_{h-1}^{(i)})}\mathbf{1}(s_{h-1}^{(i)} = s_{h-1})\mathbf{e}_{s_h^{(i)}}\bigg|\text{Data}_{h-1}\right]\mathbf{1}(E_{h-1})\widetilde{d}_{h-1}^{\pi}(s_{h-1})^2\right]$$

$$= \mathbb{E}\left[\sum_{s_{h-1}} \frac{1}{n_{s_{h-1}}}\text{Cov}\left[\frac{\pi(a_{h-1}^{(1)}|s_{h-1})}{\mu(a_{h-1}^{(1)}|s_{h-1})}\mathbf{e}_{s_h^{(1)}}\bigg|s_{h-1}^{(1)} = s_{h-1}\right]\mathbf{1}(E_{h-1})\widetilde{d}_{h-1}^{\pi}(s_{h-1})^2\right]$$

$$= \sum_{s_{h-1}}\left\{\mathbb{E}\left[\frac{1}{n_{s_{h-1}}}\mathbf{1}(E_{h-1})\widetilde{d}_{h-1}^{\pi}(s_{h-1})^2\right]\left(\sum_{a_{h-1}} \frac{\pi(a_{h-1}|s_{h-1})^2}{\mu(a_{h-1}|s_{h-1})}\text{diag}[\mathbb{P}_{h,h-1}(\cdot|s_{h-1},a_{h-1})]\right.\right.$$

$$\left.\left. - \mathbb{P}_{h,h-1}^{\pi}(\cdot|s_{h-1})[\mathbb{P}_{h,h-1}^{\pi}(\cdot|s_{h-1})]^T\right)\right\}$$

$$\prec \sum_{s_{h-1}}\left\{\frac{d_{h-1}^{\pi}(s_{h-1})^2 + \text{Var}[\widetilde{d}_{h-1}^{\pi}(s_{h-1})]}{nd_{h-1}^{\mu}(s_{h-1})(1-\delta)}\sum_{a_{h-1}} \frac{\pi(a_{h-1}|s_{h-1})^2}{\mu(a_{h-1}|s_{h-1})}\text{diag}[\mathbb{P}_{h,h-1}(\cdot|s_{h-1},a_{h-1})]\right\}$$

$$\text{(B.5)}$$

The second line uses the fact that $(s_h^{(i)}, a_h^{(i)})$ are i.i.d over $i$ given $s_{h-1}^{(i)} = s_{h-1}$. The third line uses law of total variance over $a_{h-1}^{(1)}$ as follows

$$\text{Cov}\left[\frac{\pi(a_{h-1}^{(1)}|s_{h-1})}{\mu(a_{h-1}^{(1)}|s_{h-1})}\mathbf{e}_{s_h^{(1)}}\bigg|s_{h-1}^{(1)} = s_{h-1}\right]$$

$$= \mathbb{E}\left[\left(\frac{\pi(a_{h-1}^{(1)}|s_{h-1})}{\mu(a_{h-1}^{(1)}|s_{h-1})}\right)^2\text{Cov}\left[\mathbf{e}_{s_h^{(1)}}\bigg|a_{h-1}^{(1)}, s_{h-1}^{(1)} = s_{h-1}\right]\bigg|s_{h-1}^{(1)} = s_{h-1}\right]$$

$$+ \text{Cov}\left[\frac{\pi(a_{h-1}^{(1)}|s_{h-1})}{\mu(a_{h-1}^{(1)}|s_{h-1})}\mathbb{E}\left[\mathbf{e}_{s_h^{(1)}}\bigg|a_{h-1}^{(1)}, s_{h-1}^{(1)} = s_{h-1}\right]\bigg|s_{h-1}^{(1)} = s_{h-1}\right]$$

$$= \sum_{a_{h-1}} \frac{\pi(a_{h-1}|s_{h-1})^2}{\mu(a_{h-1}|s_{h-1})}\left[\text{diag}(\mathbb{P}_{h,h-1}(\cdot|s_{h-1},a_{h-1})) - \mathbb{P}_{h,h-1}(\cdot|s_{h-1},a_{h-1})\mathbb{P}(\cdot|s_{h-1},a_{h-1})^T\right]$$

$$+ \sum_{a_{h-1}} \frac{\pi(a_{h-1}|s_{h-1})^2}{\mu(a_{h-1}|s_{h-1})}\mathbb{P}_{h,h-1}(\cdot|s_{h-1},a_{h-1})\mathbb{P}_{h,h-1}(\cdot|s_{h-1},a_{h-1})^T - \mathbb{P}_{h,h-1}^{\pi}(\cdot|s_{h-1})[\mathbb{P}_{h,h-1}^{\pi}(\cdot|s_{h-1})]^T$$

$$= \sum_{a_{h-1}} \frac{\pi(a_{h-1}|s_{h-1})^2}{\mu(a_{h-1}|s_{h-1})}\text{diag}(\mathbb{P}_{h,h-1}(\cdot|s_{h-1},a_{h-1})) - \mathbb{P}_{h,h-1}^{\pi}(\cdot|s_{h-1})[\mathbb{P}_{h,h-1}^{\pi}(\cdot|s_{h-1})]^T$$

The last line (B.5) follows from the fact that $\mathbb{P}_{h,h-1}^{\pi}(\cdot|s_{h-1})[\mathbb{P}_{h,h-1}^{\pi}(\cdot|s_{h-1})]^T$ is positive semidefinite and that $\mathbb{E}[X^2] = \text{Var}[X] + (\mathbb{E}[X])^2$. Combining (B.4) and (B.5) and by recursively apply them, we get the stated results. $\qquad\square$

Combine Lemma B.1, (B.3) and Theorem B.1 with an appropriately chosen $\delta$, we get our final result:

**Theorem 4.1** (Main Theorem, restated). *Let the immediate expected reward, its variance and the value function be defined as follows (for all $h = 1, 2, 3, ..., H$):*

$$r_h(s_h, a_h, s_{h+1}) := \mathbb{E}_{\pi}\left[r_h^{(1)}\bigg|s_h^{(1)} = s_h, a_h^{(1)} = a_h, s_{h+1}^{(1)} = s_{h+1}\right] \in [0, R_{\max}]$$

$$\sigma_h(s_h, a_h, s_{h+1}) := \text{Var}_{\pi}\left[r_h^{(1)}\bigg|s_h^{(1)} = s_h, a_h^{(1)} = a_h, s_{h+1}^{(1)} = s_{h+1}\right]^{1/2} \leq \sigma$$

$$V_h^{\pi}(s_h) := \mathbb{E}_{\pi}\left[\sum_{t=h}^{H} r_t(s_t^{(1)}, a_t^{(1)})\bigg|s_h^{(1)} = s_h\right] \in [0, V_{\max}].$$

*For the simplicity of the statement, define boundary conditions:* $r_0(s_0) \equiv 0, \sigma_0(s_0, a_0) \equiv 0, \frac{d_0^\pi(s_0)}{d_0^\mu(s_0)} \equiv$
$1, \frac{\pi(a_0|s_0)}{\mu(a_0|s_0)} \equiv 1$ *and* $V_{H+1}^\pi \equiv 0$. *Moreover, let* $\tau_a := \max_{t,s_t,a_t} \frac{\pi(a_t|s_t)}{\mu(a_t|s_t)}$ *and* $\tau_s := \max_{t,s_t} \frac{d_t^\pi(s_t)}{d_t^\mu(s_t)}$. *If the number of episodes* $n$ *obeys that*

$$n > \max\left\{ \frac{4t\tau_a\tau_s}{\min_{t,s_t}\max\{d_t^\pi(s_t), d_t^\mu(s_t)\}}, \frac{16\log n}{\min_{t,s_t} d_t^\mu(s_t)} \right\}$$

*for all* $t = 2, ..., H$, *then the our estimator* $\widehat{v}_{\mathrm{MIS}}^\pi$ *with an additional clipping step obeys that*

$$\mathbb{E}[(\mathcal{P}\widehat{v}_{\mathrm{MIS}}^\pi - v^\pi)^2] \leq \frac{1}{n}\sum_{h=0}^{H}\sum_{s_h} \frac{d_h^\pi(s_h)^2}{d_h^\mu(s_h)}\mathrm{Var}\left[\frac{\pi(a_h^{(1)}|s_h)}{\mu(a_h^{(1)}|s_h)}(V_{h+1}^\pi(s_{h+1}^{(1)}) + r_h^{(1)})\Bigg| s_h^{(1)} = s_h\right]$$

$$\cdot\left(1 + \sqrt{\frac{16\log n}{n\min_{t,s_t} d_t^\mu(s_t)}}\right) + \frac{19\tau_a^2\tau_s^2 S H^2(\sigma^2 + R_{\max}^2 + V_{\max}^2)}{n^2}.$$

*Proof of Theorem 4.1.* Choose $\delta = \sqrt{4\log(n)/(n\min_{t,s_t} d_t^\mu(s_t))}$. Lemma B.2, Lemma B.3 and Theorem B.1 provide an MSE bound of the fictitious estimator and then by substituting the resulting bound to Lemma B.1, we obtain:

$$\mathbb{E}[(\mathcal{P}\widehat{v}_{\mathrm{MIS}}^\pi - v^\pi)^2]$$

$$\leq \frac{\mathrm{Var}[V_1^\pi(s_1^{(1)})]}{n} + \frac{(1-\delta)^{-1}}{n}\sum_{h=1}^{H}\sum_{s_h} \frac{d_h^\pi(s_h)^2}{d_h^\mu(s_h)}\mathrm{Var}\left[\frac{\pi(a_h^{(1)}|s_h)}{\mu(a_h^{(1)}|s_h)}(V_{h+1}^\pi(s_{h+1}^{(1)}) + r_h^{(1)})\Bigg| s_h^{(1)} = s_h\right]$$

$$+ \frac{(1-\delta)^{-1}}{n}\sum_{h=1}^{H}\sum_{s_h} \frac{2(1-\delta)^{-1}h\tau_a\tau_s}{n}\frac{d_h^\pi(s_h)}{d_h^\mu(s_h)}\mathrm{Var}\left[\frac{\pi(a_h^{(1)}|s_h)}{\mu(a_h^{(1)}|s_h)}(V_{h+1}^\pi(s_{h+1}^{(1)}) + r_h^{(1)})\Bigg| s_h^{(1)} = s_h\right]$$

$$\tag{B.6}$$

$$+ \frac{3}{n^2}HSV_{\max}^2.$$

The first assumption on $n$ ensures that $\delta < 1/2$, which allows us to write $(1-\delta)^{-1} \leq (1+2\delta)$ in the leading term and $(1-\delta)^{-1} \leq 2$ in the subsequent terms. The second assumption on $n$ ensures that we can apply Theorem B.1 with parameter $\delta < 1/2$.

Then to obtain the simplified expression as stated in the theorem, we simply bound $d_h^\pi(s_h)/d_h^\mu(s_h) \leq \tau_s$ in (B.6), and then use the following bound

$$\mathrm{Var}\left[\frac{\pi(a_h^{(1)}|s_h)}{\mu(a_h^{(1)}|s_h)}(V_{h+1}^\pi(s_{h+1}^{(1)}) + r_h^{(1)})\Bigg| s_h^{(1)} = s_h\right]$$

$$= \mathbb{E}\mathrm{Var}\left[\frac{\pi(a_h^{(1)}|s_h)}{\mu(a_h^{(1)}|s_h)}(V_{h+1}^\pi(s_{h+1}^{(1)}) + r_h^{(1)})\Bigg| s_h^{(1)} = s_h, a_h^{(1)}, s_{h+1}^{(1)}\right]$$

$$+ \mathrm{Var}\left[\frac{\pi(a_h^{(1)}|s_h)}{\mu(a_h^{(1)}|s_h)}(V_{h+1}^\pi(s_{h+1}^{(1)}) + r_h(s_h, a_{h+1}^{(1)}, s_{h+1}^{(1)}))\Bigg| s_h^{(1)} = s_h\right]$$

$$\leq \mathbb{E}_\pi\left[\frac{\pi(a_h^{(1)}|s_h)^2}{\mu(a_h^{(1)}|s_h)^2}\Bigg| s_h^{(1)} = s_h\right]\sigma^2 + \mathrm{Var}_\mu\left[\frac{\pi(a_h^{(1)}|s_h)}{\mu(a_h^{(1)}|s_h)}(V_{h+1}^\pi(s_{h+1}^{(1)}) + r_h(s_h, a_{h+1}^{(1)}, s_{h+1}^{(1)}))\Bigg| s_h^{(1)} = s_h\right]$$

$$\leq \mathbb{E}_\pi\left[\frac{\pi(a_h^{(1)}|s_h)}{\mu(a_h^{(1)}|s_h)}\Bigg| s_h^{(1)} = s_h\right]\sigma^2 + \mathbb{E}_\pi\left[\frac{\pi(a_h^{(1)}|s_h)}{\mu(a_h^{(1)}|s_h)}(V_{h+1}^\pi(s_{h+1}^{(1)}) + r_h(s_h, a_{h+1}^{(1)}, s_{h+1}^{(1)}))^2\Bigg| s_h^{(1)} = s_h\right]$$

$$\leq \tau_a(\sigma^2 + 2V_{\max}^2 + 2R_{\max}^2).$$

The second line uses the law of total expectation, the third line replaces the variance with an upper bound $\sigma^2$, the fourth line uses $\mathrm{Var}[X] \leq \mathbb{E}[X^2]$ and a change of measure from $\mu$ to $\pi$. The last line takes the upper bound $\tau_a$, $R_{\max}$ and $V_{\max}$.

The proof is complete by combining the bounds of the second and the third term. $\square$

*Proof of Corollary 1.* The results in Corollary 1 requires a slightly different bound of (B.6) then the one we derived above. We use the assumption on $n$ to ensure that

$$\frac{4h\tau_a\tau_s}{n}\frac{d_h^\pi(s_h)}{d_h^\mu(s_h)} \leq \frac{d_h^\pi(s_h)\max\{d_h^\pi(s_h),d_h^\mu(s_h)\}}{d_h^\mu(s_h)} \leq \frac{d_h^\pi(s_h)^2}{d_h^\mu(s_h)} + d_h^\pi(s_h),$$

which gives us an upper bound of proportional to $n^{-1}H(\tau_a\tau_s + \tau_a)(\sigma^2 + H^2\mathbb{R}_{\max}^2)$. $\qquad\square$

**Remark 2** (Sample complexity in the finite action case)**.** *The result implies a sample complexity upper bound (in terms of the number of episodes) of $H^3SA/\epsilon^2$ for evaluating a fixed target policy by running an exploration policy that visits every state and action pair with probability $\Omega(1/(SA))$.*

*The Cramer-Rao lower bound for the discrete DAG-MDP model Jiang and Li [2016, Theorem 3] implies a lower bound of $H^2SA/\epsilon^2$, which suggests that our bound is optimal up to a factor of $H$ even for the cases where $A$ is small. In the settings where $A$ is unbounded. Based on our insight with the contextual bandits setting[Wang et al., 2017], we conjecture that the additional dependence on $H$ in our $H^3\tau_a\tau_s/\epsilon^2$ bound is required.*

*The comparison with the CR lower bound is a lot more delicate and interesting. We defer more detailed discussion on that to Remark 4.*

**Remark 3** (When $\pi \approx \mu$)**.** *It is not entirely straightforward to see how Theorem 4.1 gives a $H^2/n$ bound in the case of $\pi \approx \mu$ rather than the $H^3/n$ bound that we describe in Corollary 1. We make it explicit here in this remark. First the variance term in the bound can be expanded using $\mathrm{Var}[X] = \mathbb{E}[X^2] - \mathbb{E}[X]^2$.*

$$\sum_{s_h}\frac{d^\pi(s_h)^2}{d^\mu(s_h)}\mathrm{Var}\left[\frac{\pi(a_h^{(1)}|s_h)}{\mu(a_h^{(1)}|s_h)}(V_{h+1}^\pi(s_{h+1}^{(1)}) + r_h^{(1)})\bigg| s_h^{(1)} = s_h\right]$$

$$=\sum_{s_h}\frac{d^\pi(s_h)^2}{d^\mu(s_h)}\sum_{a_h}\frac{\pi(a_h|s_h)^2}{\mu(a_h|s_h)}\left(\mathbb{E}[V_{h+1}^\pi(s_{h+1})^2 + r_h(s_h,a_h,s_h')^2 + \sigma^2(s_h,a_h,s_h')|s_h,a_h]\right.$$

$$\left. + 2\mathbb{E}[V_{h+1}^\pi(s_{h+1})r_h(s_h,a_h,s_h')|s_h,a_h]\right) - \sum_{s_h}\frac{d^\pi(s_h)^2}{d^\mu(s_h)}V_h^\pi(s_h)^2$$

$$=\sum_{s_h,a_h,s_{h+1}}\frac{d^\pi(s_h,a_h,s_{h+1})^2}{d^\mu(s_h,a_h,s_{h+1})}\left(V_{h+1}^\pi(s_{h+1})^2 + [r_h^2 + \sigma_h^2 + 2r_hV_{h+1}^\pi](s_h,a_h,s_{h+1})\right) - \sum_{s_h}\frac{d^\pi(s_h)^2}{d^\mu(s_h)}V_h^\pi(s_h)^2.$$

*If we substitute the above bound into Theorem 4.1, we can see that the negative part of the bound getting combined with $\sum_{s_{h-1},a_{h-1},s_h}\frac{d^\pi(s_{h-1},a_{h-1},s_h)^2}{d^\mu(s_h,a_h,s_{h+1})}V_h^\pi(s_h)^2$ from the previous time point, which gives the following more interpretable upper bound of the leading term below*

$$\frac{1}{n}\sum_{h=0}^{H}\left[\sum_{s_{h+1}}\left(\sum_{s_h,a_h}\frac{d^\pi(s_h,a_h,s_{h+1})^2}{d^\mu(s_h,a_h,s_{h+1})} - \frac{d^\pi(s_{h+1})^2}{d^\mu(s_{h+1})}\right)V_{h+1}^\pi(s_{h+1})^2\right.$$

$$\left. + \sum_{s_h,a_h,s_{h+1}}\frac{d^\pi(s_h,a_h,s_{h+1})^2}{d^\mu(s_h,a_h,s_{h+1})}\left([r_h^2 + \sigma_h^2 + 2r_hV_{h+1}^\pi](s_h,a_h,s_{h+1})\right)\right].$$

*When $\pi = \mu$, the first term goes away and the above can be bounded by*

$$\frac{1}{n}\sum_{h=0}^{H}\sum_{s_h,a_h,s_{h+1}}d^\pi(s_h,a_h,s_{h+1})(R_{\max}r_h + \sigma^2 + 2V_1^\pi r_h) \leq \frac{1}{n}(R_{\max}V_1^\pi + H\sigma^2 + 2[V_1^\pi]^2) \leq \frac{3V_{\max}^2 + H\sigma^2}{n}.$$

*Check that when $\pi$ and $\mu$ are sufficiently close such that $\sum_{s_{h+1}}\left(\sum_{s_h,a_h}\frac{d^\pi(s_h,a_h,s_{h+1})^2}{d^\mu(s_h,a_h,s_{h+1})} - \frac{d^\pi(s_{h+1})^2}{d^\mu(s_{h+1})}\right) = O(1/H)$, then we get the same $H^2/n$ rate as above.*

**Remark 4** (Comparison to the Cramer-Rao lower bound)**.** *Theorem 3 in [Jiang and Li, 2016, Appendix C.] provides a Cramer-Rao lower bound on the variance of any unbiased estimator for a*

*simplified setting of an nonstationary episodic MDP where a reward only appear at the end of the episode and the reward is deterministic (i.e.,$\sigma^2 = 0$). Their bound, in our notation, translates into*

$$\lim_{n \to \infty} \mathrm{Var}\big[\sqrt{n}(\widehat{v}^\pi - v^\pi)\big] \geq \sum_{t=0}^{H} \mathbb{E}_\mu \left[ \frac{d^\pi(s_t^{(1)})^2}{d^\mu(s_t^{(1)})^2} \frac{\pi(a_t^{(1)}|s_t^{(1)})^2}{\mu(a_t^{(1)}|s_t^{(1)})^2} \mathrm{Var}\Big[V_{t+1}^\pi(s_{t+1}^{(1)})\Big|s_t^{(1)}, a_t^{(1)}\Big] \right].$$

*Our Theorem 4.1 implies*

$$\lim_{n \to \infty} n\mathbb{E}[(\mathcal{P}\widehat{v}_{\mathrm{MIS}}^\pi - v^\pi)^2] \leq \sum_{t=0}^{H} \mathbb{E}_\mu \left[ \frac{d^\pi(s_t^{(1)})^2}{d^\mu(s_t^{(1)})^2} \mathrm{Var}_\mu\Big[\frac{\pi(a_t^{(1)}|s_t^{(1)})}{\mu(a_t^{(1)}|s_t^{(1)})} V_{t+1}^\pi(s_{t+1}^{(1)})\Big|s_t^{(1)}\Big] \right].$$

*The upper and lower bounds are clearly very similar, with the only difference in where the importance weights of the actions are. We can verify that the upper bound is bigger because*

$$\mathrm{Var}_\mu\Big[\frac{\pi(a_t^{(1)}|s_t^{(1)})}{\mu(a_t^{(1)}|s_t^{(1)})} V_{t+1}^\pi(s_{t+1}^{(1)})\Big|s_t^{(1)}\Big]$$

$$=\mathbb{E}_\mu \left[ \mathrm{Var}\Big[\frac{\pi(a_t^{(1)}|s_t^{(1)})}{\mu(a_t^{(1)}|s_t^{(1)})} V_{t+1}^\pi(s_{t+1}^{(1)})\Big|s_t^{(1)}, a_t^{(1)}\Big]\Big|s_t^{(1)} \right] + \mathrm{Var}_\mu \left[ \mathbb{E}\Big[\frac{\pi(a_t^{(1)}|s_t^{(1)})}{\mu(a_t^{(1)}|s_t^{(1)})} V_{t+1}^\pi(s_{t+1}^{(1)})\Big|s_t^{(1)}, a_t^{(1)}\Big]\Big|s_t^{(1)} \right]$$

$$=\mathbb{E}_\mu \left[ \frac{\pi(a_t^{(1)}|s_t^{(1)})^2}{\mu(a_t^{(1)}|s_t^{(1)})^2} \mathrm{Var}\Big[V_{t+1}^\pi(s_{t+1}^{(1)})\Big|s_t^{(1)}, a_t^{(1)}\Big]\Big|s_t^{(1)} \right] + \mathrm{Var}_\mu \left[ \frac{\pi(a_t^{(1)}|s_t^{(1)})}{\mu(a_t^{(1)}|s_t^{(1)})} \mathbb{E}\Big[V_{t+1}^\pi(s_{t+1}^{(1)})\Big|s_t^{(1)}, a_t^{(1)}\Big]\Big|s_t^{(1)} \right]$$

$$=\mathbb{E}_\mu \left[ \frac{\pi(a_t^{(1)}|s_t^{(1)})^2}{\mu(a_t^{(1)}|s_t^{(1)})^2} \mathrm{Var}\Big[V_{t+1}^\pi(s_{t+1}^{(1)})\Big|s_t^{(1)}, a_t^{(1)}\Big]\Big|s_t^{(1)} \right] + \mathrm{Var}_\mu \left[ \frac{\pi(a_t^{(1)}|s_t^{(1)})}{\mu(a_t^{(1)}|s_t^{(1)})} Q_t^\pi(s_t^{(1)}, a_t^{(1)})\Big|s_t^{(1)} \right].$$

*Provided that the second term is comparable to the first, then our upper bound is rate-optimal. Both terms can be bounded by $H^2 R_{\max}^2 \mathbb{E}_\mu[\frac{\pi(a_t^{(1)}|s_t^{(1)})^2}{\mu(a_t^{(1)}|s_t^{(1)})^2}]$ and the bound cannot be improved. However, if we consider the overall bounds that sum over the $H$ items, the summation of the first term (the lower bound) is at most $H^2 \tau_a \tau_s R_{\max}^2$ (note that, somewhat surprisingly, no additional factors of $H$ is incurred), while the second term can be as large as $H^3 R_{\max}^2 \mathbb{E}_\mu[\frac{\pi(a_t^{(1)}|s_t^{(1)})^2}{\mu(a_t^{(1)}|s_t^{(1)})^2}]$ in some cases. One trivial example of that would be an MDP that gives a constant immediate reward of $R_{\max}/2$ for all $t = H/2 + 1, H/2 + 2, ..., H$. Note that in this case, $Q_t^\pi(s_t^{(1)}, a_t^{(1)}) \equiv (H - t)R_{\max}/2$, which ensures that the second term is lower bounded by*

$$\frac{1}{16} H^2 R_{\max}^2 \mathrm{Var}_\mu \left[ \frac{\pi(a_t^{(1)}|s_t^{(1)})}{\mu(a_t^{(1)}|s_t^{(1)})}\Big|s_t^{(1)} \right] = \frac{1}{16} H^2 R_{\max}^2 \left( \mathbb{E}_\mu \left[ \frac{\pi(a_t^{(1)}|s_t^{(1)})^2}{\mu(a_t^{(1)}|s_t^{(1)})^2}\Big|s_t^{(1)} \right] - 1 \right)$$

*for all $t, s_t^{(1)}$. As we sum over $t$, this leads to an $H^3$ term in our upper bound that does not exist in the Cramer-Rao lower bound.*

*A curious theoretical question is whether such an additional factor of $H$ in the error bound is required for off-policy evaluation in the small $\mathcal{S}$, large $\mathcal{A}$ setting that we considered.*

## C  Application to Other IS-Based Estimators

In this section, we discuss the applications of our marginalized approach to other IS-based estimators. We first unify some popular IS-based estimators, such as importance sampling and weighted doubly robust estimators, using a generic framework of IS-based estimators. Then we show the corresponding marginalized IS-based estimators, and provide the asymptotic unbiasedness and consistency results. At last, we provide details about how to deal with partial observability when applying our marginalized approach.

## C.1 Generic IS-Based Estimators Setup

The IS-based estimators usually provide an unbiased or consistent estimate of the value of target policy $\pi$ [Thomas, 2015]. We first provide a generic framework of IS-based estimators, and analyze the similarity and difference between different IS-based estimators. This framework could give us insight into the design of IS-based estimators, and is useful to understand the limitation of them.

Let $\rho_t^{(i)} := \frac{\pi(a_t^{(i)}|s_t^{(i)})}{\mu(a_t^{(i)}|s_t^{(i)})}$ be the importance ratio at time step $t$ of $i$-th trajectory, and $\rho_{0:t}^{(i)} :=$ $\prod_{t'=0}^{t} \frac{\pi(a_{t'}^{(i)}|s_{t'}^{(i)})}{\mu(a_{t'}^{(i)}|s_{t'}^{(i)})}$ be the cumulative importance ratio for the $i$-th trajectory. We also use $\rho_t(s_t, a_t)$ to denote $\pi(a_t|s_t)/\mu(a_t|s_t)$ over this paper. The generic framework of IS-based estimators can be expressed as follows

$$\widehat{v}^\pi = \frac{1}{n} \sum_{i=1}^{n} g(s_0^{(i)}) + \sum_{i=1}^{n} \sum_{t=0}^{H-1} \frac{\rho_{0:t}^{(i)}}{\phi_t(\rho_{0:t}^{(1:n)})} (r_t^{(i)} + f_t(s_t^{(i)}, a_t^{(i)}, s_{t+1}^{(i)})), \tag{C.1}$$

where $\phi_t : \mathbb{R}_+^n \to \mathbb{R}_+$ are the "self-normalization" functions for $\rho_{0:t}^{(i)}$, $g : \mathcal{S} \to \mathbb{R}$ and $f_t : \mathcal{S} \times \mathcal{A} \times \mathcal{S} \to \mathbb{R}$ are the "value-related" functions. Note $\mathbb{E} f_t = 0$. For the unbiased IS-based estimators, it usually has $\phi_t(\rho_{0:t}^{(1:n)}) = n$, and we first observe that the importance sampling (IS) estimator [Precup et al., 2000] falls in this framework using:

$$\text{(IS)} : \quad \begin{aligned} & g(s_0^{(i)}) = 0; \; \phi_t(\rho_{0:t}^{(1:n)}) = n; \\ & f_t(s_t^{(i)}, a_t^{(i)}, s_{t+1}^{(i)}) = 0. \end{aligned}$$

For the doubly tobust (DR) estimator [Jiang and Li, 2016], the normalization function and value-related functions are:

$$\text{(DR)} : \quad \begin{aligned} & g(s_0^{(i)}) = \widehat{V}^\pi(s_0); \; \phi_t(\rho_{0:t}^{(1:n)}) = n; \\ & f_t(s_t^{(i)}, a_t^{(i)}, s_{t+1}^{(i)}) = -\widehat{Q}^\pi(s_t^{(i)}, a_t^{(i)}) + \widehat{V}^\pi(s_{t+1}^{(i)}). \end{aligned}$$

Self-normalized estimators such as weighted importance sampling (WIS) and weighted doubly robust (WDR) estimators [Thomas and Brunskill, 2016] are popular consistent estimators to achieve better bias-variance trade-off. The critical difference of consistent self-normalized estimators is to use $\sum_{j=1}^{n} \rho_{0:t}^{(j)}$ as normalization function $\phi_t$ rather than $n$. Thus, the WIS estimator is using the following normalization and value-related functions:

$$\text{(WIS)} : \quad \begin{aligned} & g(s_0^{(i)}) = 0; \; \phi_t(\rho_{0:t}^{(1:n)}) = \sum_{j=1}^{n} \rho_{0:t}^{(j)}; \\ & f_t(s_t^{(i)}, a_t^{(i)}, s_{t+1}^{(i)}) = 0, \end{aligned}$$

and the WDR estimator:

$$\text{(WDR)} : \quad \begin{aligned} & g(s_0^{(i)}) = \widehat{V}^\pi(s_0); \; \phi_t(\rho_{0:t}^{(1:n)}) = \sum_{j=1}^{n} \rho_{0:t}^{(j)}; \\ & f_t(s_t^{(i)}, a_t^{(i)}, s_{t+1}^{(i)}) = -\widehat{Q}^\pi(s_t^{(i)}, a_t^{(i)}) + \widehat{V}^\pi(s_{t+1}^{(i)}). \end{aligned}$$

Note that, the DR estimator reduced the variance from the stochasticity of action by using the technique of control variate $f_t(s_t^{(i)}, a_t^{(i)}, s_{t+1}^{(i)})$ in value-related function, and the WDR estimators reducing variance by the bias-variance trade-off using self-normalization, especially in the presence of weight clipping [Bottou et al., 2013]. However, both could still suffer large variance, because the cumulative importance ratio $\rho_{0:t}^{(i)}$ always appear directly in this framework, which makes the variance to increase exponentially as the horizon goes long.

## C.2 Marginalized IS-Based Estimators

Recall the marginalized IS estimators (2.2), we obtain a generic framework of marginalized IS-based estimators as:

$$\widehat{v}_M(\pi) = \frac{1}{n} \sum_{i=1}^{n} g(s_0^{(i)}) + \frac{1}{n} \sum_{i=1}^{n} \sum_{t=0}^{H-1} \widehat{w}_t(s_t^{(i)}) \rho_t^{(i)} (r_t^{(i)} + f_t(s_t^{(i)}, a_t^{(i)}, s_{t+1}^{(i)})). \tag{C.2}$$

Note that the "self-normalization" function $\phi$ has not appeared in the framework above is because we can implement the self-normalization within the estimate of $w_t(s)$. Thus, the marginalized IS-based estimators can be obtained by applying different $g$ and $f_t$ in Section C.1 into framework (C.2).

We first show the equivalence between framework (C.1) and framework (C.2) in expectation if $\phi_t(\rho_{0:t}^{(1:n)}) = n$ and $\widehat{w}_t(s) = w_t(s)$.

**Lemma C.1.** *If* $\phi_t(\rho_{0:t}^{(1:n)}) = n$ *in framework* (C.1) *and* $\widehat{w}_t(s) = w_t(s)$ *in framework* (C.2)*, then these two frameworks are equal in expectation, i.e.,*

$$
\mathbb{E}\left[w_t(s_t^{(i)})\rho_t^{(i)}(r_t^{(i)} + f_t(s_t^{(i)}, a_t^{(i)}, s_{t+1}^{(i)}))\right]
$$
$$
=\mathbb{E}\left[\rho_{0:t}^{(i)}(r_t^{(i)} + f_t(s_t^{(i)}, a_t^{(i)}, s_{t+1}^{(i)}))\right]
$$

*holds for all $i$ and $t$.*

*Proof of Lemma C.1.* Given the conditional independence in the Markov property, we have

$$
\mathbb{E}\left[\rho_{0:t}^{(i)}(r_t^{(i)} + f_t(s_t^{(i)}, a_t^{(i)}, s_{t+1}^{(i)}))\right] =\mathbb{E}\left[\mathbb{E}\left[\rho_{0:t}^{(i)}(r_t^{(i)} + f_t(s_t^{(i)}, a_t^{(i)}, s_{t+1}^{(i)}))|s_t^{(i)}\right]\right]
$$
$$
=\mathbb{E}\left[\mathbb{E}\left[\rho_{0:t-1}^{(i)}|s_t^{(i)}\right]\mathbb{E}\left[\rho_t^{(i)}(r_t^{(i)} + f_t(s_t^{(i)}, a_t^{(i)}, s_{t+1}^{(i)}))|s_t^{(i)}\right]\right]
$$
$$
=\mathbb{E}\left[w_t(s_t^{(i)})\mathbb{E}\left[\rho_t^{(i)}(r_t^{(i)} + f_t(s_t^{(i)}, a_t^{(i)}, s_{t+1}^{(i)}))|s_t^{(i)}\right]\right]
$$
$$
=\mathbb{E}\left[w_t(s_t^{(i)})\rho_t^{(i)}(r_t^{(i)} + f_t(s_t^{(i)}, a_t^{(i)}, s_{t+1}^{(i)}))\right],
$$

where the first equation follows from the law of total expectation, the second equation follows from the conditional independence from the Markov property. This completes the proof. $\square$

Next, we show that if we have an unbiased or consistent estimate $\widehat{w}_t$ of $w_t$, the IS-based OPE estimators that simply replace $\prod_{t'=0}^{t-1} \frac{\pi(a_{t'}|s_{t'})}{\mu(a_{t'}|s_{t'})}$ with $\widehat{w}_t(s_t)$ will remain unbiased or consistent.

**Theorem C.1.** *Let* $\phi_t(\rho_{0:t}^{(1:n)}) = n$ *in framework* (C.1)*, then framework* (C.2) *could keep the unbiasedness and consistency same as in framework* (C.1) *if $\widehat{w}_t(s)$ is an unbiased or consistent estimator for marginalized ratio $w_t(s)$ for all $t$:*

1. *If an unbiased estimator falls in framework* (C.1)*, then its marginalized estimator in framework* (C.2) *is also an unbiased estimator of $v^\pi$ given unbiased estimator $\widehat{w}_t(s)$ for all $t$.*

2. *If a consistent estimator falls in framework* (C.1)*, then its marginalized estimator in framework* (C.2) *is also a consistent estimator of $v^\pi$ given consistent estimator $\widehat{w}_t(s)$ for all $t$.*

*Proof of Theorem C.1.* We first provide the proof of the first part of unbiasedness. Given $\mathbb{E}[\widehat{w}_t^n(s)|s] = w_t(s)$ for all $t$, then

$$
\mathbb{E}\left[\widehat{w}_t^n(s_t^{(i)})\rho_t^{(i)}(r_t^{(i)} + f_t(s_t^{(i)}, a_t^{(i)}, s_{t+1}^{(i)}))\right] =\mathbb{E}\left[\mathbb{E}\left[\widehat{w}_t^n(s_t^{(i)})\rho_t^{(i)}(r_t^{(i)} + f_t(s_t^{(i)}, a_t^{(i)}, s_{t+1}^{(i)}))|s_t^{(i)}\right]\right]
$$

$$
=\mathbb{E}\left[\mathbb{E}\left[\widehat{w}_t^n(s_t^{(i)})|s_t^{(i)}\right]\mathbb{E}\left[\rho_t^{(i)}(r_t^{(i)} + f_t(s_t^{(i)}, a_t^{(i)}, s_{t+1}^{(i)}))|s_t^{(i)}\right]\right]
$$

$$
=\mathbb{E}\left[w_t(s_t^{(i)})\mathbb{E}\left[\rho_t^{(i)}(r_t^{(i)} + f_t(s_t^{(i)}, a_t^{(i)}, s_{t+1}^{(i)}))|s_t^{(i)}\right]\right]
$$

$$
=\mathbb{E}\left[w_t(s_t^{(i)})\rho_t^{(i)}(r_t^{(i)} + f_t(s_t^{(i)}, a_t^{(i)}, s_{t+1}^{(i)}))\right]
$$
$$
=\mathbb{E}\left[\rho_{0:t}^{(i)}(r_t^{(i)} + f_t(s_t^{(i)}, a_t^{(i)}, s_{t+1}^{(i)}))\right], \tag{C.3}
$$

where the the first equation follows from the law of total expectation, the second equation follows from the conditional independence of the Markov property, the last equation follows from Lemma C.1. Since the original estimator falls in framework (C.1) is unbiased, summing (C.3) over $i$ and $t$ completes the proof of the first part.

We now prove the second part of consistency. Since we have

$$\plim_{n\to\infty} \frac{1}{n} \sum_{i=1}^{n} \sum_{t=1}^{H} \widehat{w}_t^n(s_t^{(i)}) \rho_t^{(i)}(r_t^{(i)} + f_t(s_t^{(i)}, a_t^{(i)}, s_{t+1}^{(i)})) = \sum_{t=1}^{H} \plim_{n\to\infty} \frac{1}{n} \sum_{i=1}^{n} \widehat{w}_t^n(s_t^{(i)}) \rho_t^{(i)}(r_t^{(i)} + f_t(s_t^{(i)}, a_t^{(i)}, s_{t+1}^{(i)})),$$

then, to prove the consistency, it is sufficient to show

$$\plim_{n\to\infty} \frac{1}{n} \sum_{i=1}^{n} \widehat{w}_t^n(s_t^{(i)}) \rho_t^{(i)}(r_t^{(i)} + f_t(s_t^{(i)}, a_t^{(i)}, s_{t+1}^{(i)})) = \plim_{n\to\infty} \frac{1}{n} \sum_{i=1}^{n} \rho_{0:t}^{(i)}(r_t^{(i)} + f_t(s_t^{(i)}, a_t^{(i)}, s_{t+1}^{(i)})),$$

(C.4)

given $\plim_{n\to\infty} \widehat{w}_t^n(s) = w_t(s)$ for all $s \in \mathcal{S}$. Note that $d_t^\mu(s)$ is the state distribution under behavior policy $\mu$ at time step $t$, then for the left hand side of (C.4), we have

$$\plim_{n\to\infty} \frac{1}{n} \sum_{i=1}^{n} \widehat{w}_t^n(s_t^{(i)}) \rho_t^{(i)}(r_t^{(i)} + f_t(s_t^{(i)}, a_t^{(i)}, s_{t+1}^{(i)}))$$

$$= \sum_{s\in\mathcal{S}} d_t^\mu(s) \plim_{n\to\infty} \left[ \frac{1}{n} \sum_{i=1}^{n} \widehat{w}_t^n(s) \frac{\pi(a_t^{(i)}|s)}{\mu(a_t^{(i)}|s)} \mathbf{1}(s_t^{(i)} = s)(r_t^{(i)} + f_t(s, a_t^{(i)}, s_{t+1}^{(i)})) \right]$$

$$= \sum_{s\in\mathcal{S}} d_t^\mu(s) \plim_{n\to\infty} \left[ \widehat{w}_t^n(s) \frac{1}{n} \sum_{i=1}^{n} \frac{\pi(a_t^{(i)}|s)}{\mu(a_t^{(i)}|s)} \mathbf{1}(s_t^{(i)} = s)(r_t^{(i)} + f_t(s, a_t^{(i)}, s_{t+1}^{(i)})) \right]$$

$$= \sum_{s\in\mathcal{S}} d_t^\mu(s) \left[ \plim_{n\to\infty} (\widehat{w}_t^n(s)) \plim_{n\to\infty} \left( \frac{1}{n} \sum_{i=1}^{n} \frac{\pi(a_t^{(i)}|s)}{\mu(a_t^{(i)}|s)} \mathbf{1}(s_t^{(i)} = s)(r_t^{(i)} + f_t(s, a_t^{(i)}, s_{t+1}^{(i)})) \right) \right]$$

$$= \sum_{s\in\mathcal{S}} d_t^\mu(s) w_t(s) \plim_{n\to\infty} \left[ \frac{1}{n} \sum_{i=1}^{n} \frac{\pi(a_t^{(i)}|s)}{\mu(a_t^{(i)}|s)} \mathbf{1}(s_t^{(i)} = s)(r_t^{(i)} + f_t(s, a_t^{(i)}, s_{t+1}^{(i)})) \right]$$

$$= \sum_{s\in\mathcal{S}} d_t^\mu(s) w_t(s) \mathbb{E}\left[ \frac{\pi(a_t|s)}{\mu(a_t|s)}(r_t + f_t(s, a_t, s_{t+1})) \Big| s_t = s \right],$$

(C.5)

where the first equation follows from the weak law of large number. Similarly, for the right hand side of (C.4), we have

$$\plim_{n\to\infty} \frac{1}{n} \sum_{i=1}^{n} \rho_{0:t}^{(i)}(r_t^{(i)} + f_t(s_t^{(i)}, a_t^{(i)}, s_{t+1}^{(i)}))$$

$$= \sum_{s\in\mathcal{S}} d_t^\mu(s) \plim_{n\to\infty} \left[ \frac{1}{n} \sum_{i=1}^{n} \prod_{t'=0}^{t-1} \frac{\pi(a_{t'}^{(i)}|s_{t'}^{(i)})}{\mu(a_{t'}^{(i)}|s_{t'}^{(i)})} \mathbf{1}(s_t^{(i)} = s) \frac{\pi(a_t^{(i)}|s)}{\mu(a_t^{(i)}|s)}(r_t^{(i)} + f_t(s, a_t^{(i)}, s_{t+1}^{(i)})) \right]$$

$$= \sum_{s\in\mathcal{S}} d_t^\mu(s) \mathbb{E}\left[ \prod_{t'=0}^{t-1} \frac{\pi(a_{t'}|s_{t'})}{\mu(a_{t'}|s_{t'})} \frac{\pi(a_t|s)}{\mu(a_t|s)}(r_t + f_t(s, a_t, s_{t+1})) \Big| s_t = s \right]$$

$$= \sum_{s\in\mathcal{S}} d_t^\mu(s) \mathbb{E}\left[ \prod_{t'=0}^{t-1} \frac{\pi(a_{t'}|s_{t'})}{\mu(a_{t'}|s_{t'})} \Big| s_t = s \right] \mathbb{E}\left[ \frac{\pi(a_t|s)}{\mu(a_t|s)}(r_t + f_t(s, a_t, s_{t+1})) \Big| s_t = s \right]$$

$$= \sum_{s\in\mathcal{S}} d_t^\mu(s) w_t(s) \mathbb{E}\left[ \frac{\pi(a_t|s)}{\mu(a_t|s)}(r_t + f_t(s, a_t, s_{t+1})) \Big| s_t = s \right],$$

(C.6)

where the first equation follows from the weak law of large number and the third equation follows from the conditional independence of the Markov property. Thus, we have (C.5) equal to (C.6). This completes the proof of the second half. $\qquad\square$

In partially observable MDPs (POMDPs), we may not be able to obverse all states. However, if there exist any observable states, our marginalized approach could leverage these observable states to reduce variance. That is, we use the partial trajectory from the closest observable states to the current time step to represent the current state. Assume the current time step is $t$ and the closest observable states is $s_{t-L}$ at time step $t-L$, then we can use $\frac{d_t^\pi(s_{t-L})}{d_t^\mu(s_{t-L})}\prod_{i=t-L}^{t-1}\frac{\pi(a_i|s_i)}{\mu(a_i|s_i)}$ as $w_t(s_t)$, while other IS-based methods are equivalent to using $\prod_0^{t-1}\frac{\pi(a_i|s_i)}{\mu(a_i|s_i)}$ as $w_t(s_t)$. The observable states in POMDPs can be considered as the states that can be reunioned at in the DAG MDPs. If there is no observable state in POMDPs, then it is equivalent that DAG MDPs is reduced to tree MDPs. Definition of DAG and Tree MDPs can be found in the extended version of [Jiang and Li, 2016].

Finally, we propose a new marginalized IS estimator to further improve the data efficiency and reduce variance. Since DR only reduces the variance from the stochasticity of action [Jiang and Li, 2016] and our marginalized estimator (C.2) reduce the variance from the cumulative importance weights, it is also possible to reduce the variance the stochasticity of reward function.

Based on the definition of MDPs, we know that $r_t$ is the random variable that only determined by $s_t, a_t$. Thus, if $\widehat{R}(s,a)$ is an unbiased and consistent estimator for $R(s,a)$, $r_t^{(i)}$ in framework (C.2) can be replaced by that $\widehat{R}(s_t^{(i)}, a_t^{(i)})$, and keep unbiasedness or consistency same as using $r_t^{(i)}$.

Note that we can use an unbiased and consistent Monte-Carlo based estimator

$$\widehat{r}(s_t, a_t) = \frac{\sum_{i=1}^n r_t^{(i)}\mathbf{1}(s_t^{(i)} = s_t, a_t^{(i)} = a_t)}{\sum_{i=1}^n \mathbf{1}(s_t^{(i)} = s_t, a_t^{(i)} = a_t)},$$

and then we obtain a better marginalized framework

$$\widehat{v}_{BM}(\pi) = \frac{1}{n}\sum_{i=1}^n g(s_0^{(i)}) + \frac{1}{n}\sum_{i=1}^n \sum_{t=0}^{H-1} \widehat{w}_t(s_t^{(i)})\rho_t^{(i)}(\widehat{r}(s_t^{(i)}, a_t^{(i)}) + f_t(s_t^{(i)}, a_t^{(i)}, s_{t+1}^{(i)})). \quad \text{(C.7)}$$

**Remark 5.** *Note that, the only difference between* (C.2) *and* (C.7) *is* $r_t^{(i)}$ *and* $\widehat{r}(s_t^{(i)}, a_t^{(i)})$. *Thus, the unbiasedness or consistency of* (C.7) *can be obtained similarly by following Theorem* C.1 *and its proof.*

One interesting observation is that when each $(s_t, a_t)$-pair is observed only once in $n$ iterations, then framework (C.7) reduces to (C.2). Note that when this happens, we could still potentially estimate $\widehat{w}_t^n(s_t)$ well if $|\mathcal{A}|$ is large but $|\mathcal{S}|$ is relative small, in which case we can still afford to observe each potential values of $s_t$ many times. Thus, we can also obtain better marginalized IS-based estimators, e.g., the MIS and MDR estimators we use in our experiments, by applying different $g$ and $f_t$ in Section C.1 into framework (C.7).

# D   Details of Experiments

In this section, we first clarify the experiment settings. We also provide a detailed discussion about the preference of MIS and SSD-IS. Finally, we provide the extended experiential results about applying MIS to doubly robust related approaches.

## D.1   Environment Settings

**ModelWin MDP**   As depicted in Figure 1(a), the agent in the ModelWin domain always begins in $s_1$, where it must select between two actions. The first action $a_1$ causes the agent to transition to $s_2$ with probability $p$ and $s_3$ with probability $1 - p$. The second action $a_2$ does the opposite. We set $p = 0.4$. The agent receives a reward of 1 every time the state transitions to $s_2$, $-1$ to $s_3$, and 0 otherwise.

**ModelFail MDP**   The dynamics of ModelFail MDP (Figure 1(b)) is similar to ModelWin, but the reward is delayed after the unobservable states — the agent receives a reward of 1 only when it arrives $s_1$ from the left state and $-1$ only when it arrives $s_1$ from the right state. We set $p = 1$ to make the problem easier.

Policy $\pi$ takes action $a_1$ and $a_2$ with probabilities $0.2$ and $0.8$ when at state $s_1$ or observing "?". $\mu$ take actions uniformly at random.

We remark that the partial-state observability in **ModelFail** is specialized and should be distinguished from the more general partial observability considered in the classical POMDP literature. The two distinctive (and clearly artificial) differences are that

1. There are checkpoints of full state observability every other step.

2. We assume that the action probability is logged when the observation is "?".

The model-based approach clearly fails when "?" is treated as if it is a state when building the model. A standard POMDP that uses just a memory of size 2 will resolve this issue without any trouble. One may also consider an alternative MDP that only takes the checkpoint states $s_1$ as states, but the two actions that the policies will take are no longer a function of just $s_1$ (in this example it actually is because the observation is always "?" in the step after $s_1$).

Finally, both **ModelWin** and **ModelFail** are highly specialized examples with deterministic transitions into the states $s_1$ that could potentially generate rewards for some actions. Moreover, there are no non-trivial actions involved as we transition from $s_2$ and $s_3$ back to $s_1$. This means that we can perfectly estimate the marginal state-distribution of $s_1$ with just one data point in all methods. As a result, we do not expect the results to reveal the worst-case dependence on the model parameters such as $H$. The following example fixes that.

**Non-stationary Non-mixing MDP**   In the time-varying MDP example, we consider the following carefully designed MDP where there are two states and a continuous action in $[0, 1]$. In State $0$ the agent always transitions to State $0$, regardless of the actions. In State $1$, it transitions to State $0$ deterministically if the action is taken to be within an unknown subset of measure $1/H$ within $[0, 0.5]$. This subset might be different for different $t$. When the agent is at State $0$, then a reward of $1$ is received regardless of the actions taken when the step number is larger than $H/2$; otherwise no reward is received.

The behavior and target policy (probability density on $[0, 1]$) are defined to be

$$\mu(a|s = 1) = 1 \text{ for all } a,$$

and

$$\pi(a|s = 1) = \begin{cases} 1.9 & \text{if } a \in [0, 0.5] \\ 0.1 & \text{otherwise.} \end{cases}$$

and $\pi(a|s = 0) = \mu(a|s = 0) = 1$.

This example is deliberately designed such that we have a non-stationary dynamics[6] that does not really mix beyond a constant factor so an additional factor of $H$ in the sample complexity can potentially appear. Meanwhile, the cumulative reward is proportional to $H$ for the target policy, so we expect to see a $H^1.5$ dependence in the (relative) RMSE curves as we vary $H$. Finally, due to the non-mixing property of this example, and the importance weight of stationary distributions based on SSD-IS is expected to be biased. All the above observations are consistent with what we see in the experiments presented in Figure 3.

**Mountain Car**   Mountain Car domain is a classic control problem with 2-dimensional state space (position and velocity), 3 discrete one-dimensional actions (push left, no push, push right), and deterministic dynamics. We follow the same dynamic as in [Sutton and Barto, 1998]. The horizon, $H$, is set to be 100. We use initial state distribution to be uniform in position and $0$ in velocity to ensure exploratory. Since our proposed method mainly focuses on the tabular setting, we use the state aggregation for both MIS and SSD-IS to ensure fair comparison: position is multiplied by $2^6$ and velocity is multiplied by $2^8$, and then we use the rounded integers to be the abstract state (adopted from [Jiang and Li, 2016]). Thus, the (marginalized or stationary) state distribution can be estimated on the tabular abstract states.

## D.2 Detailed Discussions

The ModelWin domain is only a very special case of episodic fully-observable MDPs. Even if we use the stationary state distribution (estimated by ignoring the within-episode step count in the dataset) instead of marginalized state distribution in (3.1), that value will still happen to be correct in both time-invariant and time-varying case. However, that is not correct in general. As the results we showed in the Mountain Car domain, SSD-IS fails to provide correct evaluation. That is because SSD-IS is designed for the infinite-horizon problems and usually cannot be directly applied to the episodic problems, where SSD-IS uses the stationary distribution ($t \to \infty$) to approximate that for all $t = 1, ..., H$ which is biased and not consistent even as the number of episodes $n \to \infty$ in general. For example, in the Mountain Car domain, the stationary state distribution, $\prod_{t=1}^{\infty} P_{s_{t-1},s_t}^{\pi} d_0$, will converge to the probability mass on the absorbing state with any exploratory policy $\pi$.

The result of mountain car experiment in the current version is slightly different from the early version. There are two main modifications in that experiments: 1. In the early version, the on-policy estimated $v^{\pi}$ for calculating RMSE did not use enough trajectories, so that the curves in the early version are biased. 2. We changed the implementation of SSD-IS in our current version. Previously, we solved Equation (8) and (9) in [Liu et al., 2018a] using an iterative approach. The current implementation solves Equation (8) and (9) in [Liu et al., 2018a] directly by re-formalizing it to be a quadratic programming problem. The current implementation follows the released code provided by the author of [Liu et al., 2018a], and the detailed description of that can be found in Appendix D.3.

We also explain the reason of MIS outperforming MDR in Figure 5 and Figure 6. In the MDR methods, we split our dataset into two halves. We use one half to estimate the marginalized state distribution, and the other half to estimate the Q-function. Intuitively, since Q-function is only used to be a control variant in the estimator, we suppose the statistical error from the marginalized state distribution may dominate the overall statistical error. As MIS uses the whole data to estimate the marginalized state, whereas MDR only uses a half data, the statistical error of MIS could be lass that of MDR. The theoretical explanation of that goes beyond the topic of this paper, and we will leave it as the future work.

## D.3 SSD-IS with finite state space.

The pioneering work by Liu et al. [2018a] describes a method — SSD-IS — for estimating the ratio of stationary state distribution under $\pi$ and $\mu$ for an infinite horizon (possibly discounted) MDP.

The estimator is described primarily for the case when the state is a continuous variable, which requires defining a reproducing kernel Hilbert space (RKHS) and solving a mini-max problem.

To be a bit more self-contained in this paper, we provide the concise formula using our notation for estimating the stationary distribution $d_\infty^\pi(s)$ as well as directly estimating the importance ratio

$$\rho(s) := \frac{d_\infty^\pi(s)}{d_{1:N-1}^\mu(s)}$$

between the $d_\infty^\pi(s)$ and the marginalized state distribution under $\mu$ that measures the average state-visitation within the first $N$ iterations (note that we can observe triplets $(s_t, a_t, s_{t+1})$ for all $t = 1, ..., N-1$).

Note that a roll-out in an infinite-horizon environment of a fixed length $N$ can be denoted in our notation with a single episode $n = 1$ and horizon $H = N$.

The master equation that we need is the following:

$$d_\infty^\pi(s') = P^\pi(s'|s)d_\infty^\pi(s) = P^\pi(s'|s)d_{1:H-1}^\mu(s)\frac{d_\infty^\pi(s)}{d_{1:H-1}^\mu(s)},$$

which, in matrix form, is:

$$d_\infty^\pi = A^{\pi,\mu}\mathrm{Diag}(d_{1:H-1}^\mu)^{-1}d_\infty^\pi \tag{D.1}$$

where $A^{\pi,\mu} \in \mathbb{R}^{S \times S}$ and $A^{\pi,\mu}(s', s)$ measures the joint distribution of $s \sim d_t^\mu$ with a randomly chosen $t$ from $1, ..., H-1$ and $s'$ that is obtained by taking an action according to $\pi$ at $s$.

By left-multiplying $\mathrm{Diag}(d_{1:H-1}^\mu)^{-1}$ on both sides of the equation, we also get

$$\rho = \mathrm{Diag}(d_{1:H-1}^\mu)^{-1}A^{\pi,\mu}\rho. \tag{D.2}$$

Observe that (D.1) and (D.2) are eigenvalue problems of $d_\infty^\pi$ and $\rho$. They differ only by whether we normalize the $A^{\pi,\mu}$ matrix on the left or on the right by multiplying the diagonal matrix $\mathrm{Diag}(d_{1:H-1}^\mu)^{-1}$.

They suggest that if we can consistently estimate $A^{\pi,\mu}$ and $\mathrm{Diag}(d_{1:H-1}^\mu)^{-1}$, then **the right eigenvector of the corresponding estimated matrices with eigenvalue closest to** 1 will be consistent estimators of $d_\infty^\pi$ and $\rho$.

Note that we can estimate the joint-distribution $A^{\pi,\mu}[s', s]$ by importance sampling using

$$\hat{A}^{\pi,\mu}[s', s] = \frac{1}{n} \sum_{i=1}^{n} \frac{1}{H-1} \sum_{t=1}^{H-1} \frac{\pi(a_t^{(i)}|s_t^{(i)})}{\mu(a_t^{(i)}|s_t^{(i)})} \mathbf{1}(s_t^{(i)} = s, s_{t+1}^{(i)} = s')$$

with potentially infinite action. And $d_{1:H-1}^\mu$ can be estimated by

$$\hat{d}_{1:H-1}^\mu(s) = \frac{1}{n} \sum_{i=1}^{n} \frac{1}{H-1} \sum_{t=1}^{H-1} \mathbf{1}(s_t^{(i)} = s).$$

For the infinite horizon case, we can just take $n = 1$.

We emphasize that while the above results and the spectral estimators were not explicitly presented by Liu et al. [2018a], they are simply a rewriting of Equation (8) and (9) in [Liu et al., 2018a] more explicitly in a more specialized case.

The SSD-IS implementation that we used in the experiments with discrete state space corresponds to this particular version that we described in this section, which is the same as the version of the code released by the authors modulo some boundary conditions[7]. These boundary conditions seem to be important for getting SSD-IS to work correctly for the finite horizon MDPs.

That said, we acknowledge that when the underlying MDP is stationary and $H$ is large enough relative to the mixing rate of the MDP, then using the estimated importance weight $\rho$ to construct importance sampling estimators as in SSD-IS may provide a favorable bias-variance trade-off in finite sample, because its variance is smaller by a factor of $H$ than the standard MIS while its bias on the estimated importance ratio $\frac{d_t^\pi(s)}{d_t^\mu(s)}$ decays exponentially as $t$ gets larger.

### D.4 Extended Experimental Studies

We now present further empirical results. To test the use of our approach in other IS-based estimators, we compared DR, WDR, MDR, and MIS in the same environments, where DR denotes the doubly robust estimator [Jiang and Li, 2016], WDR denotes the weighted doubly robust estimator [Thomas and Brunskill, 2016], MIS denotes the estimator using proposed marginalized approach used with doubly robust, and MIS is our marginalized importance sampling estimator. The estimates of $d_t^\pi$ and $d_t^\mu$ are projected to the probability simplex in our MDR and MIS estimators. The results are obtained in the same environments as Section 5.

The results are in Figure 5 and Figure 6. These demonstrate that other IS based methods can also leverage our marginalized approach to benefit performance dramatically.

## E  Algorithm Details

Algorithm 1 summarizes our method of marginalized off-policy evaluation. Note that the MIS estimator in Section 5 is using the estimate of $d_t^\pi(\cdot)$ by normalizing (E.1) into the probability simplex for better performance.

(a) ModelWin MDP with different number of episodes, $n$ (b) ModelWin MDP with different horizon, $H$ (c) ModelFail MDP with different number of episodes, $n$ (d) ModelFail MDP with different horizon, $H$

Figure 5: Results on Time-invariant MDPs.

Figure 6: Mountain Car with different number of episodes.

---

**Algorithm 1** Marginalized Off-Policy Evaluation

---

**Input:** Transition data $\mathcal{D} = \{\{s_t^{(i)}, a_t^{(i)}, r_t^{(i)}, s_{t+1}^{(i)}\}_{t=0}^{H-1}\}_{i=1}^{n}$ from the behavior policy $\mu$. A target policy $\pi$ which we want to evaluate its cumulative reward.

1: Calculate the on-policy estimation of $d_0(\cdot)$ by

$$\widehat{d}_0(s) = \frac{1}{n}\sum_{i=1}^{n}\mathbf{1}(s_0^{(i)} = s),$$

and set $\widehat{d}_0^{\mu}(\cdot)$ and $\widehat{d}_0^{\pi}(\cdot)$ as $\widehat{d}_0(s)$.

2: **for** $t = 0, 1, \ldots, H-1$ **do**

3:     Choose all transition data as time step $t$, $\{s_t^{(i)}, a_t^{(i)}, r_t^{(i)}, s_{t+1}^{(i)}\}_{i=1}^{n}$.

4:     Calculate the on-policy estimation of $d_{t+1}^{\mu}(\cdot)$ by

$$\widehat{d}_{t+1}^{\mu}(s) = \frac{1}{n}\sum_{i=1}^{n}\mathbf{1}(s_{t+1}^{(i)} = s).$$

Calculate the off-policy estimation of $d_{t+1}^{\pi}(\cdot)$ by

$$\widehat{d}_{t+1}^{\pi}(s) = \frac{1}{n}\sum_{i=1}^{n}\frac{\widehat{d}_t^{\pi}(s_t^{(i)})}{\widehat{d}_t^{\mu}(s_t^{(i)})}\frac{\pi(a_t^{(i)}|s_t^{(i)})}{\mu(a_t^{(i)}|s_t^{(i)})}\mathbf{1}(s_{t+1}^{(i)} = s) \tag{E.1}$$

5:     Estimate the reward function

$$\widehat{r}(s_t, a_t) = \frac{\sum_{i=1}^{n} r_t^i \mathbf{1}(s_t^i = s_t, a_t^i = a_t)}{\sum_{i=1}^{n}\mathbf{1}(s_t^i = s_t, a_t^i = a_t)}.$$

6:     Normalize $d_{t+1}^{\pi}(\cdot)$ into the probability simplex, and specify $\widehat{w}_{t+1}(s)$ as $\dfrac{\widehat{d}_{t+1}^{\pi}(s)}{\widehat{d}_{t+1}^{\mu}(s)}$ for each $s$.

7: **end for**

8: Substitute the all estimated values above into (C.7) to obtain $\widehat{v}(\pi)$, the estimated cumulative reward of $\pi$.

---