[Reviews · NeurIPS 2019]

Reviewer 1



Originality: The main idea of the paper - avoiding the long horizon problem by computing IS over state distributions rather than trajectories - was already introduced in (Liu et. al. 2018), which the authors cite sufficiently often in the text. However, the approach the authors take to leveraging this idea is original. Additionally, there is not yet enough published work on leveraging this potentially important idea (IS over state distribution), and therefore even being the second paper in this direction is still charting new territory. Quality - To the extent I looked at it the theoretical work is solid. I did not go over every equality in the proofs to check for algebraic errors, but I did go through every step in the proofs found in the appendix. I have some comments on the exposition of the proof, but I leave that to the "clarity" section. The experimental section is passable (apart from a potential problem which I will detail later), but somewhat below what I'd expect of a publication in NeurIPS. Considering the ModelWin/ModelFail domains as toy examples for demonstration, the only actual domain is the mountain car which is a fairly simple example. I think the paper should definitely try to include some more domains. Even other classical control tasks such as acrobot, cartpole, etc.. I expect these shouldn't be hard to implement if the authors already have the mountain car implementation. A specific baseline I would really like to see the authors add is the PDIS (per-decision IS) and CWPDIS (consistent weighted per-decision IS). The reason I would like to see these specific estimators is that I think they would be interesting for comparison in the ModelWin/ModelFail domains in which every two steps (or one, depending if you count the actionless transition back to s_1 a step) effectively start a new trajectory. I am guessing that is why the MIS which needs to observe a lot of transitions, not trajectories does so well for the increasing horizon experiment. However, the PDIS estimator might be able to somewhat benefit from this property, albeit not as much, since it does not have the problem of non-per-decision methods for which decisions different form the evaluation policy late in the trajectory diminish the utility of "good" observed transitions early in the trajectory. I think the most interesting comparison in the experimental section should be made with the SSD-IS estimator since they both utilize the same general philosophy, and I think the authors should discuss it more thoroughly. Why does it achieve EXACTLY the same results as DM for model fail? Why does it perform as well as MIS for mountain car but eventually stops improving? (is that a limitation of the function class used as discriminator functions in the implementation of SSD-IS?). Lastly there is one critical question I have regarding the time-varying MDP example, but that may be just a misunderstanding on my part. If p_t is sampled uniformly at each time step, isn't the probability of transition (for example from s_1 to s_2 given a_1) p(p_t)*p_t, and marginalizing over p_t make that setting equivalent to a time-invariant MDP with p=3.5? Clarity: Overall the paper is clear but could be improved. The motivation and and background is clear. Section 4 (theory) is clear, but section 4.1 which attempts to sketch the proof is confusing. The proof as it is written in the appendix is well written, but the authors should do a better job of sketching it in the main text. Sketching proofs is always hard but I'd prefer if the authors sketched less steps, and limit themselves to the bare basics which would be stated clearly in the main text with proper references to the (well written) appendix. Apart from the confusion regarding the time-varying MDP case which I hope the authors would clarify, the experimental section is well written but as I mentioned should have better discussion of comparison with SSD-IS. Less important, but the description of the ModelWin/ModelFail case is needlessly confusing and can be stripped to an easier to grasp explanation if in the ModelWin case the agent also gets the reward upon transition from s_{2 or 3} to s_1. This would not change any of the dynamics but would reduce the difference between the models to full vs. partial observability without introducing the needless difference in reward timing. It should also be stated that the agent needs to choose an action at states s_{2 or 3}, but it doesn't matter which one it chooses. Significance - I think the paper is fairly significant. Post review update - The authors addressed some of my concerns regarding the clarity of the paper, and I have revised my score accordingly. Regarding the authors' explanation of the comparison with SSD-IS - I find the authors' explanation plausible, but not entirely convincing. Despite the SSD-IS making the assumption that the state distribution is stationary, I encountered cases where in practice SSD-IS performs well even if that assumption is broken. If the authors are correct, I think they should be able to demonstrate their claims empirically. In addition to including in the main text the explanation the authors gave, I think the paper would greatly benefit from showing in the appendix the states distributions at different times, and correlating them with the error of both the SSD-IS and MIS. By showing such plots, the authors should be able to convincingly demonstrate their claim that the difference in performance is a result of the SSD-IS not working with a true stationary state distribution.

Reviewer 2



The paper presents an algorithm for finite-horizon off-policy policy evaluation based on a new way of estimating stationary distribution correction ratios called Marginalized Importance Sampling. The paper derives Marginalized Importance Sampling, gives a theoretical analysis of the algorithm's sample complexity (showing it possesses an optimal dependence on horizon), and presents strong results on simple MDPs, time-varying MDPs, and the Mountain Car domain. I recommend accepting the paper for publication. To my knowledge the method is original, the quality of the analysis seems good, the writing is clear enough, and the work seems significant and likely to be extended. EDIT: I have read the author's feedback.

Reviewer 3



Originality: From my personal knowledge I think the method is new, though the author has pointed out they borrow the related idea from [Liu et al., 2018a] that accumulate distribution over state, but the approach is different (Liu's paper mainly focus on continuous setting and use function approximation to learn a weight function, while this paper is mainly focus on tabular setting and approximate using empirical statistics). Quality: The theory part is sound with clear explanation and detailed discussion. For the experimental part, since no code is provided and I couldn't check the details, I will ask the author several questions for the comparison with other baseline, especially the SSD-IS method. The main concern is why MIS perform even better at time-invariant environment where SSD-IS should have more data to use when estimating the density ratio. To be more concrete here are two questions I would like to ask: 1. In Figure 2 and 3, why DM and SSD-IS method works well in ModelWin but perform very bad at ModelFail? For me it is surprised in time-invariant environment SSD-IS method perform worse than MIS method. Any explanation? 2. In Figure 3 (b) and (d), why the curve is not smooth even after 128 repetition? Clarity: The paper is well written and easy to read. Significance: I like the idea of the paper and OPE problem become more and more important in research area. Overall: I think the paper is a good paper but I will remain below borderline until the author answer for my experimental problems.

[Author Response · NeurIPS 2019]

We would like to thank the reviewers for appreciating our novel contributions on the algorithmic and theoretical front!
We focus on clarifying our experimental results in this rebuttal.

*[Why DM fails at ModelFail and SSD-IS achieve EXACTLY the same results as DM at ModelFail?].*

ModelFail was first introduced by Thomas and Brunskill [2016] to show the failure of model-based approach in the
MDPs with some partial observability. In ModelFail, the agent cannot tell the difference between any of the states
except for $s_1$, but both DM and SSD-IS require full observability. From the point of view of both DM and SSD-IS, the
actions have no impact on state transitions or rewards, so every policy has the same cumulative reward (equal to the the
true cumulative reward of the behavior policy). A detailed discussion about why DM fails at ModelFail can be found in
[Thomas and Brunskill, 2016, Section D.1]. MIS can handle partial observability by using observable states and the
partial trajectories between them. Please refer Section 5.1 (line 258-262, there is a typo in Line 262, $\frac{\pi(a_{2\tau}^{(i)}|s_{2\tau}^{(i)})}{\mu(a_{2\tau}^{(i)}|s_{2\tau}^{(i)})}$ should
be $\frac{\pi(a_{2\tau}^{(i)}|?)}{\mu(a_{2\tau}^{(i)}|?)}$, where symbol "?" stands for "unobserved", is an observed variable that the policy needs to react upon).
Also see Section C (line 567-575) in the supplement for more details.

*[Why MIS outperforms SSD-IS in time-invariant environments (including MountainCar) when $n$ is large?].*

The time-invariant ModelWin and MountainCar we used in the paper are finite-horizon *undiscounted* MDPs. Even
though these environments have time-invariant transitions, the state marginal distributions at each $t$ actually change
with time and only converge to the stationary distribution as $t \to \infty$.

SSD-IS uses the stationary distribution ($t \to \infty$) to approximate that for all $t = 1, ..., H$ which is biased and not
consistent even as the number of episodes $n \to \infty$. MIS, on the other hand, uses nearly unbiased and consistent
estimators of the state marginals at every $t$. This allows MIS to outperform SSD-IS on Mountain Car when $n$ gets large.
We believe this is the reason and we will investigate it in details in our future work.

## Reviewer #1

*["A specific baseline I would really like to see the authors add is the PDIS (per-decision IS) and CWPDIS (consistent*
*weighted per-decision IS)."]*

The IS and WIS in the experiments are step-wise, which are essentially PDIS and CWPDIS. The detailed explanation is
in Section 3 and Section C.

*["Why does it (SSD-IS) achieve . . . perform as well as MIS for mountain car but eventually stops improving?"]*

Please check the answers at the beginning.

*["If $p_t$ is sampled uniformly at each time step, isn't . . . setting equivalent to a time-invariant MDP with $p = 3.5$?"]*

Sorry for the confusion. Note that each transition probability $p_t$ is only sampled before the experiments and fixed during
the experiments for all episodes. We will clarify it in the final version.

## Reviewer #2

Thanks for supporting our paper. We are planning to extend our approach to large-scale environments with extensive
function approximation.

## Reviewer #3

*["In Figure 2 and 3, why DM and SSD-IS method works well in ModelWin but perform very bad at ModelFail?"]*
*["For me it is surprised in time-invariant environment SSD-IS method perform worse than MIS method."]*

Please check the answers at the beginning.

*["In Figure 3 (b) and (d), why the curve is not smooth even after 128 repetition?"]*

Note that the Y-axis is relative MSE, which is normalized by the true cumulative reward. In this time-varying MDP
(Figure 3), the true cumulative reward is related to the transition probabilities $p_t$ at each time step. We sample each
$p_t$ before the experiments and then fix them during the experiments, so the true cumulative reward is a non-smooth
function of $H$ and the figures with increasing $H$ should not be smooth. In the time-invariant MDP (Figure 2), the true
cumulative reward is a smooth function of $H$ and the corresponding figures are smooth.

## References

Thomas, P. and Brunskill, E. (2016). Data-efficient off-policy policy evaluation for reinforcement learning. In
*International Conference on Machine Learning*, pages 2139–2148.


[Meta-Review · NeurIPS 2019]

The paper studies the important problem of off-policy policy evaluation in long-horizon MDPs. The setting focuses on small-state, large-action problems. A novel estimator is proposed, whose finite-sample statistical properties are studied. Empirical results show the method is useful, especially in partially observable problems. Reviewers feel the experiment section can be strengthened (e.g., using more domains). Furthermore, the assumption that the state space is small limits the significant and applicability of this work. On the other hand, they all agree the approach is novel and useful in some cases.